# Application of machine learning models for property prediction to targeted protein degraders

Giulia Peteani[1], Minh Tam Davide Huynh [1], Grégori Gerebtzoff [1] & Raquel Rodríguez-Pérez [1]✉

Machine learning (ML) systems can model quantitative structure-property relationships (QSPR) using existing experimental data and make property predictions for new molecules. With the advent of modalities such as targeted protein degraders (TPD), the applicability of QSPR models is questioned and ML usage in TPD-centric projects remains limited. Herein, ML models are developed and evaluated for TPDs' property predictions, including passive permeability, metabolic clearance, cytochrome P450 inhibition, plasma protein binding, and lipophilicity. Interestingly, performance on TPDs is comparable to that of other modalities. Predictions for glues and heterobifunctionals often yield lower and higher errors, respectively. For permeability, CYP3A4 inhibition, and human and rat microsomal clearance, misclassification errors into high and low risk categories are lower than 4% for glues and 15% for heterobifunctionals. For all modalities, misclassification errors range from 0.8% to 8.1%. Investigated transfer learning strategies improve predictions for heterobifunctionals. This is the first comprehensive evaluation of ML for the prediction of absorption, distribution, metabolism, and excretion (ADME) and physicochemical properties of TPD molecules, including heterobifunctional and molecular glue sub-modalities. Taken together, our investigations show that ML-based QSPR models are applicable to TPDs and support ML usage for TPDs' design, to potentially accelerate drug discovery.

Machine learning (ML) models are invaluable tools for predicting the absorption, distribution, metabolism, and excretion (ADME) properties of small molecules[1–4]. For ADME predictions, ML models relate compound structural information to molecular properties, which is also referred to as quantitative structure-property relationship (QSPR) models[5]. QSPR models and ADME predictions play a pivotal role in drug discovery, assisting in the early identification of lead compounds with favorable pharmacokinetics and reduced potential for toxicity[5,6]. By accurately predicting ADME profiles, models can accelerate compound characterization and potentially reduce the costs associated with synthesis and experimental testing[7,8].

ML-based QSPR models can be created using either all available data for a certain property (global models) or smaller data sets relating to a particular discovery project or chemical series (local models)[9–12]. Local models focus on predicting ADME properties within specific chemical series or compound classes, utilizing specialized knowledge and chemical features relevant to those compounds. In contrast, global ML models capture the complex relationships between molecular structures and ADME properties across the chemical space[3,13]. As it was shown in Di Lascio et al., this broader applicability makes global models more advisable, despite the common intuition that local models might capture series- or project-specific QSPRs more accurately[2]. Therefore,

[1]Novartis Biomedical Research, Novartis Campus, 4002 Basel, Switzerland. ✉e-mail: raquel.rodriguez_perez@novartis.com

global ADME models should generally be the ones influencing prioritization and selection of lead compounds early in pharmaceutical research. However, global ML models have been predominantly utilized for the prediction of ADME properties of traditional small molecules and it is still an open question whether they are applicable to more recent drug modalities[14]. With the emergence of targeted protein degradation (TPD) as a promising therapeutic strategy, the development and evaluation of ML models for predicting molecular properties of TPDs has gained attention to assist in degraders' design[15,16]. Recent works, including a pharmaceutical industry perspective by Volak et al.[17], have highlighted the knowledge gap in how ML models perform for ADME property predictions in the TPD space.

TPDs agents represent an innovative therapeutic strategy to induce the selective degradation of disease-causing proteins through the intracellular ubiquitin-proteasome system[18]. These agents simultaneously bind the target protein and an E3 ligase, facilitating the recruitment of the protein to the cellular degradation machinery. By modulating previously 'undruggable' targets, TPD agents offer new opportunities for therapeutic intervention. Molecular glues and heterobifunctionals are two TPD submodalities. Glues are a class of molecules that directly bind to both the target protein and an E3 ubiquitin ligase, promoting the formation of a ternary complex that facilitates the ubiquitination and target degradation. Heterobifunctionals consist of a ligand to the target, a ligand to the E3 ubiquitin ligase, and a linker[19]. This complex brings the target protein in proximity to E3 ligase leading to ubiquitination and its degradation[18]. The structural features, mechanisms of action, and target engagement modes that distinguish TPDs from traditional small molecules challenge ML models' performance and generalizability in the context of TPD agents, which remain uncertain[16]. It has been recently reported that computational approaches might not be suited to TPD molecules and data limitations prevent ML-based QSPR modeling assessments[16]. Therefore, it is not yet known whether reliable predictions are possible or, in contrast, TPDs might be outside the applicability domain of ADME models[14,16]. Given the promising clinical results of recent TPDs, investigating whether ML-based QSPR models can leverage existing data to effectively predict ADME properties of TPDs is of utmost interest for pharmaceutical research[16].

Herein, ML models are generated and evaluated for the prediction of ADME properties of TPD compounds, with special focus on glues and heterobifunctionals. By leveraging on global models' generalization capability, the potential of ML models to capture the QSPR across diverse compound classes and physicochemical and ADME properties is investigated. Predictive performance of property predictions for glues and heterobifunctionals is compared and put in context to all compound modalities. Moreover, transfer learning techniques are adopted with the aim of refining ML models and improving predictions on TPD compounds.

## Results
### Assay data and global models
A data set with twenty-five ADME endpoints was utilized for ML modeling. ML-based property predictions were carried out with global QSPR models, which learn from all available data for a given ADME property or assay[2]. Here, four multi-task (MT) global models were generated to predict related properties or assays[2,20,21]. This algorithm was selected because MT learning enables the modeling of multiple properties, assays or, more generally, prediction tasks simultaneously[22–24]. The assays or tasks included in the four global MT models were:

**Permeability model** (5-task model): Apparent permeability ($P_{app}$) from low-efflux MDCK (LE-MDCK) permeability assay (versions 1 and 2), PAMPA and Caco-2 permeability assay, and efflux ratio from MDCK-MDR1 permeability assay.

**Clearance model** (6-task model): Intrinsic clearance ($CL_{int}$) from CYP metabolic stability in liver microsomes assays for rat, human, mouse, dog, cynomolgus monkey, and minipig[20].

**Binding/Lipophilicity model** (10-task model): Plasma protein binding (PPB) for rat, human, mouse, dog and cynomolgus monkey, human serum albumin (HSA) binding, microsomal binding, brain binding, and octanol-water partition and distribution coefficients (LogP and LogD).

**Cytochrome P450 (CYP) inhibition model** (4-task model): time-dependent inhibition of CYP3A4 and reversible inhibition of CYP3A4, CYP2C9, and CYP2D6.

These models are ensembles of a message-passing neural network (MPNN) coupled with a feed-forward deep neural network (DNN)[25,26]. More details can be found in the Methods section.

Due to data availability, prospective evaluation was done for a subset of endpoints. Table 1 lists fifteen physicochemical and ADME assays and properties that were considered for models' evaluation. Experiments for molecules registered until the end of 2021 were used for model generation, whereas performance was evaluated with the most recent ADME experiments, following a temporal validation.

TPDs belonging to the submodalities of glues and heterobifunctionals were identified in the data set. Global models' performance was assessed for glue and heterobifunctional TPDs separately. Figure 1 reports the number of training and test compounds across all modalities, for heterobifunctionals, and glues. For all endpoints, TPD compounds constitute less than 6% than the rest of drug modalities. Supplementary Fig. 1 shows the distribution of assay values for each modality.

Figure 2 characterizes the data set distribution and chemical space per each compound modality. Figure 2A shows the distribution of calculated descriptors used in the Lipinski's rule of five (Ro5) for glues, heterobifunctionals, and the rest of compounds in the test set. Those calculated descriptors include molecular weight (MW), hydrogen bond acceptors (HBA), hydrogen bond donors (HBD), topological polar surface area (TPSA), calculated LogP (cLogP), and number of rotatable bonds. Heterobifunctional TPDs have a larger molecular weight than the glues and are always beyond the Ro5 (bRo5). The rest of compounds tested on these ADME assays, which can belong to different drug modalities, have a molecular weight distribution more similar to that of glues. The percentage of compounds bRo5 is 19% for glues, and 34% for the rest of modalities. Since ML models have been traditionally applied to compounds with molecular weight lower than 900 or 1000 Da, and mostly for compounds following the Ro5, one could anticipate that heterobifunctional TPDs might be outside the applicability of those standard ML-based QSPR models. Calculated properties' distributions (MW, HBA, HBD, TPSA, cLogP, and rotatable bonds) are also reported for the training and test sets in Supplementary Fig. 2.

Figure 2B reports a chemical space representation based on Uniform Manifold Approximation and Projection (UMAP), which shows the distribution of TPDs and compounds from other modalities utilizing Tanimoto as the distance metric and MACCS (Molecular ACCess System) keys[27] as molecular representation. The UMAP illustrates that chemical spaces of TPDs and the rest of the compounds in the test data set only partly overlap, and TPD compounds tend to cluster together. Clusters of heterobifunctional TPDs overlap with glue compounds.

### Prediction errors on TPDs and all modalities
First, model performance was assessed for properties with at least five compounds in the test sets. Figure 3A reports the mean absolute error (MAE) for fifteen ADME endpoints, in order of increasing model error (across all modalities). This figure shows the prediction error estimations for glue and heterobifunctional TPDs separately, as well as the

**Table 1 | Assays, models, and prediction tasks**

| Property (Short name) | Assay/Property | Model | Regression task | Low risk | High risk |
|---|---|---|---|---|---|
| LE-MDCK v2 $P_{app}$ | Low efflux MDCK low efflux (LE-MDCK permeability assay (version 2). Apparent permeability ($P_{app}$) A to B | Permeability | $LogP_{app}$ | $P_{app} > 5$ cm$^{-6}$/s | $P_{app} \leq 1.5$ cm$^{-6}$/s |
| RLM $CL_{int}$ | CYP metabolic stability in rat liver microsomes (RLM). Intrinsic clearance ($CL_{int}$) | Clearance | $LogCL_{int}$ | $CL_{int} \leq 100$ µL min$^{-1}$ mg$^{-1}$ | $CL_{int} > 300$ µL min$^{-1}$ mg$^{-1}$ |
| HLM $CL_{int}$ | CYP metabolic stability in human liver microsomes (HLM). $CL_{int}$ | Clearance | $LogCL_{int}$ | $CL_{int} \leq 100$ µL min$^{-1}$ mg$^{-1}$ | $CL_{int} > 300$ µL min$^{-1}$ mg$^{-1}$ |
| MLM $CL_{int}$ | CYP metabolic stability in mouse liver microsomes (MLM). $CL_{int}$ | Clearance | $LogCL_{int}$ | - | - |
| DLM $CL_{int}$ | CYP metabolic stability in dog liver microsomes (DLM). $CL_{int}$ | Clearance | $LogCL_{int}$ | - | - |
| CyLM $CL_{int}$ | CYP metabolic stability in cynomolgus monkey liver microsomes (CyLM). $CL_{int}$ | Clearance | $LogCL_{int}$ | - | - |
| rPPB | Rat plasma protein binding (PPB) measured in multiple assays. Fraction unbound in plasma ($F_{u,p}$) | Binding/ Lipophilicity | $LogF_u$ | - | - |
| hPPB | Human PPB measured in multiple assays. $F_{u,p}$ | Binding/ Lipophilicity | $LogF_u$ | - | - |
| cynoPPB | Cynomolgus monkey PPB measured in multiple assays. $F_u$ | Binding/ Lipophilicity | $LogF_u$ | - | - |
| LogP | Rapid-throughput octanol-buffer lipophilicity measurement based on shake flask equilibrium and LC/MSMS. Partition coefficient, LogP | Binding/ Lipophilicity | LogP | - | - |
| LogD | Rapid-throughput octanol-buffer lipophilicity measurement based on shake flask equilibrium and LC/MSMS. Distribution coefficient, LogD | Binding/ Lipophilicity | LogD | - | - |
| CYP3A4 $k_{obs}$ | CYP3A4 Time dependent inhibition: Enzyme half-life. Inactivation rate ($k_{obs}$) | CYP inhibition | $Logk_{obs}$ | $k_{obs} < 0.01$ min$^{-1}$ | $k_{obs} > 0.025$ min$^{-1}$ |
| CYP3A4 $IC_{50}$ | CYP3A4 (Midazolam) inhibition: HLM. Half-maximal inhibitory concentration ($IC_{50}$) | CYP inhibition | $pIC_{50}$ | $IC_{50} \geq 10$ µM | $IC_{50} < 1$ µM |
| CYP2C9 $IC_{50}$ | CYP2C9 (Diclofenac) inhibition: HLM. $IC_{50}$ | CYP inhibition | $pIC_{50}$ | - | - |
| CYP2D6 $IC_{50}$ | CYP2D6 (Bufuralol) inhibition: HLM. $IC_{50}$ | CYP inhibition | $pIC_{50}$ | - | - |

Reported are the model and prediction tasks under evaluation. Classification thresholds to categorize numerical predictions of the regression task into low, medium, and high risk categories are shown. Low and high risk thresholds are also used to categorize experimental assay read-outs.

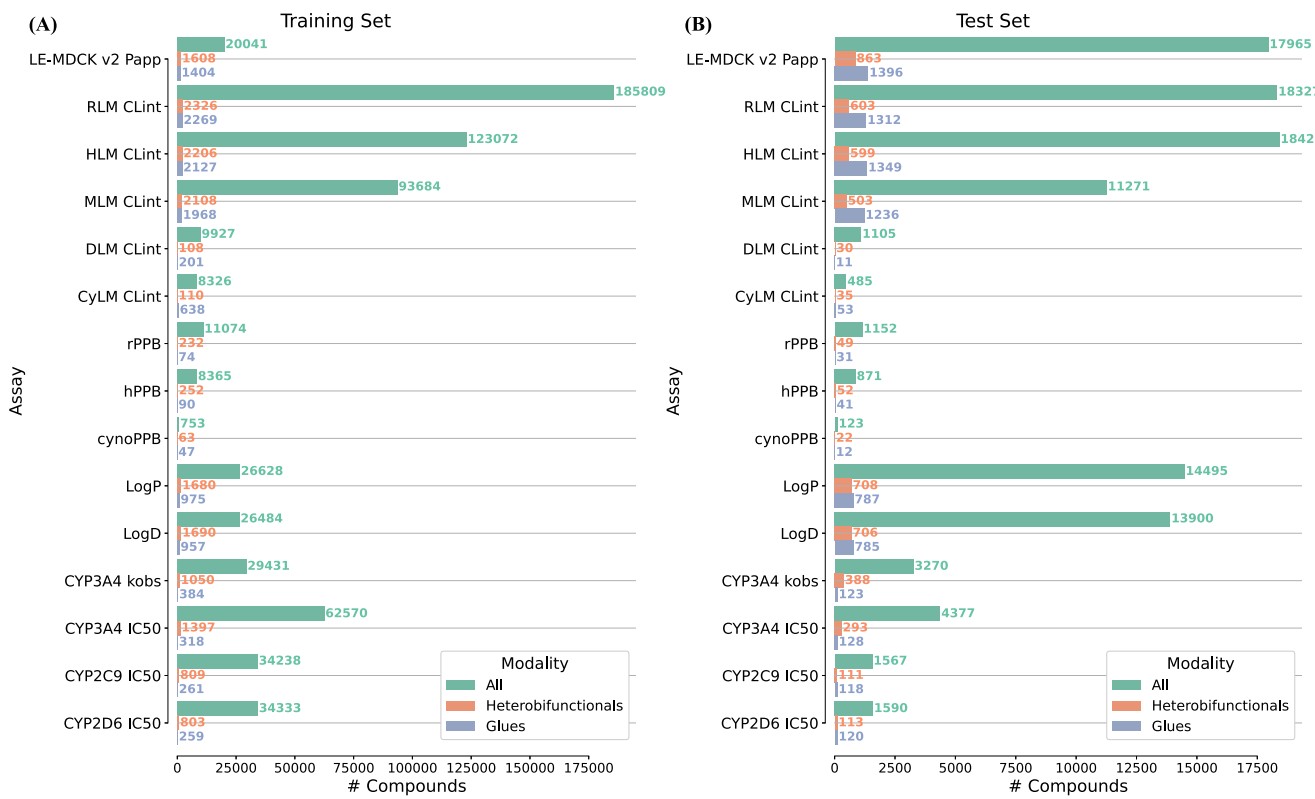

**Fig. 1 | Training and test set statistics.** The number of compounds per assay is reported both for the (**A**) training and (**B**) test sets. Shown are the number of compounds across all modalities (green), heterobifunctionals (orange), and glues (blue). Assays are described in Table 1. Source data are provided as a Source Data file.

average errors for the rest of compounds. As a control, models' errors were compared to a baseline predictor. For all test compounds, the baseline model gave a constant prediction value which corresponded to the mean property value in the training set. Such baseline prediction was consistently less accurate than ML predictions for any of the modalities. MAE values for the baseline model ranged from 0.28 (for CYP2C9 IC$_{50}$) to 0.96 (for LogD). In contrast, the largest MAE values for ML models were 0.33 for the test set with all compound modalities (LogD), 0.39 for the heterobifunctionals' test set (LE-MDCK v2 P$_{app}$), and 0.31 for the glues' test set (rPPB). Therefore, predictions were consistently lower than ~2-fold for glues and all modalities' compounds. For heterobifunctionals, average prediction errors were smaller than 2.5-fold across all studied ADME properties. The largest differences between average ML errors and baseline predictor errors were observed for lipophilicity (ΔMAE = 0.63 for LogD and ΔMAE = 0.57 for LogP). For cynomolgus monkey clearance predictions (CynLM CL$_{int}$), there was also a large error difference between ML and baseline predictor (ΔMAE = 0.33). For CYP3A4 TDI, ML-based predictions were closer to the control baseline (ΔMAE = 0.05), which highlights limited predictive ability for k$_{obs}$ values.

To further assess predictive performance, Fig. 3B shows a distribution of average errors for each endpoint on bootstrap samples ($n$ = 1000). This analysis also helps to incorporate the uncertainty of the errors' estimation due to the small sample size in some of the data sets. Similar trends were obtained both in Fig. 3A, B. Overall, results show consistency between the ML models' performance on all modalities and TPDs. Even though some property predictions were less accurate for heterobifunctional TPDs, model errors were not consistently larger than those for other modalities. Perhaps surprisingly, for the majority of the considered ADME assays, glue molecules had predictions with the lowest errors. For the bootstrap results, at least 75% of the glues had predictions with lower errors than the other molecules in the test set (either heterobifunctionals or all modalities)

for nine out of fifteen properties (cynoPPB, CYP3A4 IC$_{50}$, LE-MDCK v2 P$_{app}$, CYP3A4 k$_{obs}$, RLM CL$_{int}$, LogP, MLM CL$_{int}$, HLM CL$_{int}$, LogD). This is illustrated by the third quartiles of glues' MAE distributions in Fig. 3B. Depending on the property to predict, models had larger or smaller errors in TPDs or other modalities, but overall results suggest that TPDs are inside the domain of applicability of ML-based QSPR models.

## Performance evaluation for larger data sets

For the five ADME endpoints with most data points available, a detailed evaluation was carried out. Apart from the regression predictions, performance was estimated for categorical predictions. A compound with an experimental readout in a medium range could likely be assigned to another category if the experiment was repeated. Therefore, three classes are often utilized to categorize experimental results into high and low risk, while incorporating experimental variability. Similarly, focusing predictions on the extremes of the distribution, higher agreement with the experimental readout is ensured (higher precision). Thus, medium-range predictions (between the low and high thresholds) were set to 'inconclusive (medium)' to avoid making decisions based on those predictions. This approach helps flagging low-confidence predictions. Property thresholds are defined in Table 1.

Table 2 reports the number of test compounds, MAE values, misclassification errors for the low and high classes, and percentage of inconclusive (medium) predictions. Importantly, LE-MDCK permeability, TDI and reversible inhibition of CYP3A4, and CL$_{int}$ in human and rat liver microsomes were predicted with errors lower than 2-fold (MAE < 0.3). Figure 4 shows the classification predictions for these five selected assays in all modalities, heterobifunctional and glue TPDs. Results show better models' performance on glues than heterobifunctional TPDs. Moreover, misclassification errors were at most 14.5% and always lower than 4% for glues. These results indicate that when models provide a high or low classification prediction, there is a high confidence that it is correct. The percentage of 'inconclusive

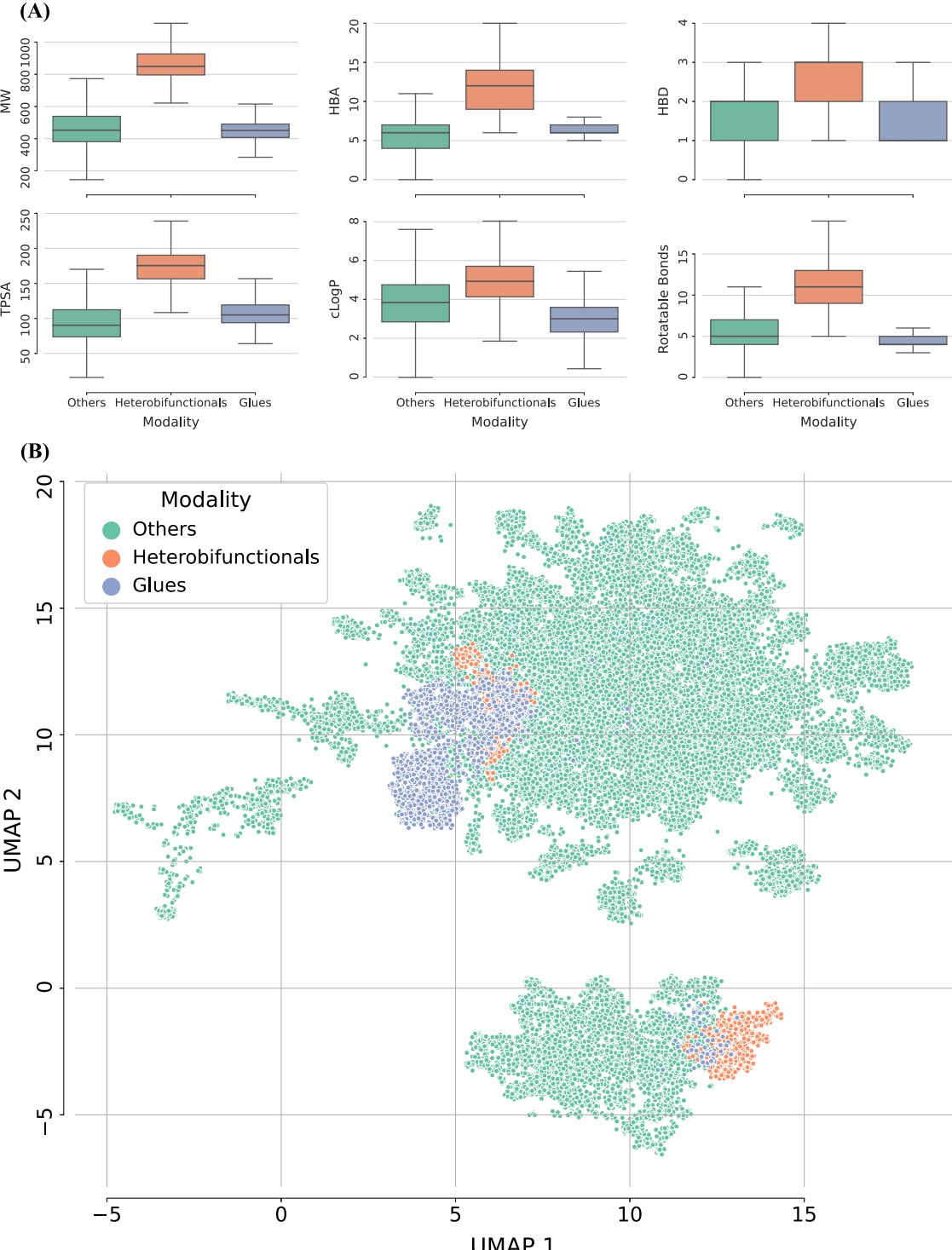

**Fig. 2 | Distribution of calculated properties and chemical space representation. A** The distributions of molecular weight (MW), number of hydrogen bond acceptors (HBA) and donors (HBD), topological polar surface area (TPSA), calculated LogP (cLogP), and number of rotational bonds are reported for glues ($n_{glues}$ = 1851, blue), heterobifunctionals ($n_{heterobifunctionals}$ = 1064, orange), and all the rest of modalities ($n_{other\ modalities}$ = 28886, green). Boxplots show the median (center line), and 1st and 3rd quartiles (Q1 and Q3, respectively). The error bars correspond to the Q1-(1.5*IQR) and Q3 + (1.5*IQR) range (IQR = Inter-Quartile Range). Datapoints below Q1 − (1.5*IQR) or above Q3 + (1.5*IQR) are considered outliers and not shown in the boxplots. **B** A Uniform Manifold Approximation and Projection (UMAP) based on Tanimoto distance and MACCS keys is shown per modality (glues, heterobifunctionals, and others) and for the test set ($n_{glues}$ = 1851, $n_{heterobifunctionals}$ = 1064, $n_{other\ modalities}$ = 28886). Source data are provided as a Source Data file.

(medium)' predictions was generally less than 35% and it was also lower for glues than heterobifunctional TPDs. Specifically, the percentage of inconclusive (medium) predictions ranged from 2% to 20% for glues, and from 19% to 51% for heterobifunctional TPDs. Supplementary

Table 1 reports the class distributions according to the assay values. There are also experimental measurements that fall into the medium category and ranged from 7% (CYP2C9 IC$_{50}$) to 45% (hPPB) of the compounds. As highlighted above, due to experimental variability,

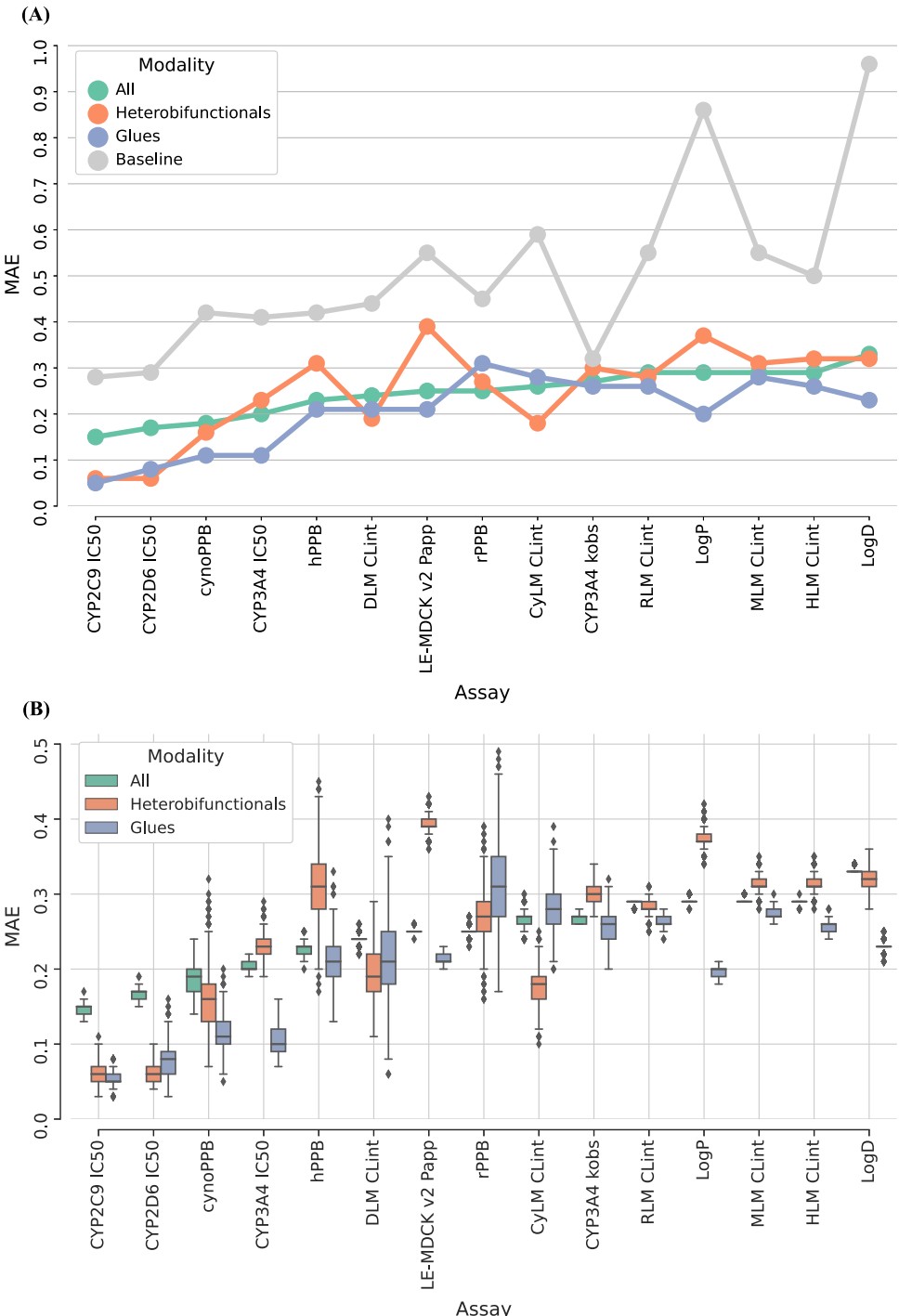

**Fig. 3 | Global models' performance on targeted protein degraders (TPDs) and other modalities.** Global machine learning (ML) model results are shown for fifteen absorption, distribution, metabolism, and excretion (ADME) assays. Reported are the mean absolute error (MAE) distributions for glues (blue), heterobifunctionals (orange) and all the other compounds (green). Results are reported for the complete test set (**A**) and bootstrap samples (n = 1000) (B). In (**A**), global models are compared to a baseline prediction (gray), i.e. mean of the training set. Boxplots in (**B**) show the median (center line), and 1st and 3rd quartiles (Q1 and Q3, respectively). The error bars correspond to the Q1-(1.5*IQR) and Q3 + (1.5*IQR) range (IQR = Inter-Quartile Range). Assays are described in Table 1. Source data are provided as a Source Data file.

such medium-range experiments could also switch category (to low or high property values) if the assay was repeated[20].

## Models' refinement for TPD compounds
Since ADME properties for heterobifunctional compounds were more challenging to predict, models' refinements were carried out with the aim of improving ADME predictions. Since data for model refinement and testing was required, investigation focused on the five properties previously evaluated: passive permeability (LE-MDCK $P_{app}$), metabolic clearance in rat liver microsomes (RLM $CL_{int}$) and human liver microsomes (HLM $CL_{int}$), CYP3A4 TDI (CYP3A4 $k_{obs}$) and reversible inhibition of CYP3A4 (CYP3A4 $IC_{50}$),

Fine-tuning strategies were investigated. Deep learning algorithms trained in one or more domains can be adapted to a different

**Table 2 | Global models' regression and classification performance for all modalities and targeted protein degraders (TPDs)**

| Property | Modality | # Test cpds | MAE | Error (Low risk class) | Error (High risk class) | Inconclusive (medium) predictions |
|---|---|---|---|---|---|---|
| LE-MDCK v2 $P_{app}$ | All modalities | 17960 | 0.25 | 3.9% | 2.6% | 14% |
| | Heterobifunctionals | 863 | 0.39 | 11.6% | 14.5% | 26% |
| | Glues | 1395 | 0.21 | 2.2% | 2.0% | 18% |
| RLM $CL_{int}$, | All modalities | 18322 | 0.29 | 3.7% | 5.0% | 25% |
| | Heterobifunctionals | 602 | 0.28 | 2.9% | 0% | 22% |
| | Glues | 1311 | 0.26 | 3.1% | 0% | 20% |
| HLM $CL_{int}$ | All modalities | 18420 | 0.29 | 3.6% | 6.4% | 20% |
| | Heterobifunctionals | 598 | 0.32 | 9.2% | 7.1% | 34% |
| | Glues | 1348 | 0.26 | 2.6% | 0% | 10% |
| CYP3A4 $k_{obs}$ | All modalities | 3,270 | 0.27 | 4.1% | 8.1% | 38% |
| | Heterobifunctionals | 388 | 0.30 | 11.1% | 2.1% | 51% |
| | Glues | 123 | 0.26 | 1.7% | 0% | 4% |
| CYP3A4 $IC_{50}$ | All modalities | 4,377 | 0.20 | 0.8% | 2.1% | 16% |
| | Heterobifunctionals | 293 | 0.23 | 1.3% | 0% | 19% |
| | Glues | 128 | 0.11 | 0.8% | 0% | 2% |

Reported are performance metrics for five absorption, distribution, metabolism, and excretion (ADME) properties that had more than 120 heterobifunctional and glue compounds (cpds) in the test sets. The number (#) of test cpds is also reported. Global models' performance is shown for regression (MAE) and classification, i.e. errors in the low and high-risk classes, and % of inconclusive (medium) predictions.

but related target domain[28,29]. This concept of transfer learning for domain adaptation was applied herein to adapt global models to a specific area of the chemical space. Existing global ML models were adapted to the TPD modality using fine-tuning with weights initialization[28,30]. Two approaches were investigated: (i) fine-tuning ML models with all new compounds registered in 2022, and (ii) fine-tuning ML models on specific chemistry (heterobifunctional TPDs registered before 2023). These two strategies are schematized in Fig. 5.

For the original MT-GNN models and the two fine-tuning approaches, Supplementary Fig. 3 reports classification predictions for permeability (LogP_app from LE-MDCK), rat and human CLint (LogCL_int), TDI of CYP3A4 (Logk_obs), and reversible inhibition of CYP3A4 (pIC_50). Moreover, Fig. 6 shows the average regression errors (MAE values) for the same models and assays. For all ADME endpoints except permeability, model fine-tuning with heterobifunctional compounds yielded equivalent or lower prediction errors than using all new data. In contrast, permeability predictions were more accurate when the MT-GNN model was fine-tuned with all new data (across all modalities) instead of heterobifunctional TPDs only. Interestingly, for all evaluated properties, fine-tuned models consistently led to lower prediction errors compared to the original MT-GNN models.

Newer experiments can also be utilized for model retraining, where the model is generated again from scratch. Global MT-GNN models were retrained with compounds registered before 2023 and tested on the same heterobifunctional molecules. Table 3 reports regression and classification prediction performance for the fine-tuned model for heterobifunctional TPDs (fine-tuning strategy 2), and the original and retrained MT-GNN global models. Results show that using most recent data for modeling consistently decreases prediction errors both in numerical property predictions and misclassifications. However, fine-tuning with heterobifunctional TPD data yielded the lowest misclassification errors across all assays and risk categories, except for low LE-MDCK permeability values. Moreover, these results indicate that when a prediction is reported by the fine-tuned ML model, it is of high confidence. Errors were lower than 4% for LE-MDCK permeability, TDI of CYP3A4, and rat CL_int, lower than 13% for human CL_int, and no errors were observed for the reversible inhibition of CYP3A4.

Hence, despite the larger efforts of model retraining (more computationally intensive and time-consuming), it did not yield performance improvements compared to fine-tuning. Even though both types of model refinements were successful in improving predictions, using a pre-trained global model and refining predictions with a relevant data set (i.e. TPD modality) can be a more promising strategy.

## Public surrogate data and ML model

Due to the recent emergence of this new therapeutic modality, there is a lack of TPD data in the public domain. This limits the possibility of generating and evaluating data-driven ML models for property prediction, especially applicable to TPDs. To accelerate ML-based QSPR for TPDs, a surrogate data set was generated and used for model building[31,32]. Public compound structures were extracted from ChEMBL[33], ZINC[34], and PROTAC-DB[35], as detailed in the Methods section, and annotated with our in-house MT-GNN models' predictions. This surrogate data set contains ~274,000 compounds with predicted data for twenty-five properties, which were included as tasks in the original MT-GNN models. The same MT-GNN approaches (equivalent architecture and hyperparameters) were trained with the surrogate data to generate new models. The code to generate the models and get predictions is provided as Supplementary Software.

The quality of the public surrogate ML models was evaluated with the same internal test set and performance was compared to the original MT-GNN models. Figure 7 shows the MAE values for the fifteen assays under evaluation for the public surrogate models. Prediction errors were estimated for glues, heterobifunctionals, and all modalities independently. As observed with the internal MT-GNN models, property predictions for heterobifunctionals were often associated to larger errors. On the other hand, average performance for glue TPDs was generally similar to the one observed across all modalities, and even higher for some assays. A control baseline was also included, where the average in the training set (in this case, predictions from the original models) was predicted for all test compounds. Such baseline often yielded higher prediction errors, but in a few cases ML-based predictions were of equivalent quality. Hence, surrogate models were not always applicable, especially to predict some properties for heterobifunctional TPDs. For instance, predictions of time-dependent inhibition of CYP3A4 (Logk_obs values) had MAE values larger than the baseline for heterobifunctional compounds.

The applicability of the original global models and surrogate models might not be equivalent due to differences in chemical space coverage and labels' quality (experiment vs. prediction). However, results suggest that surrogate models' predictions can be successful

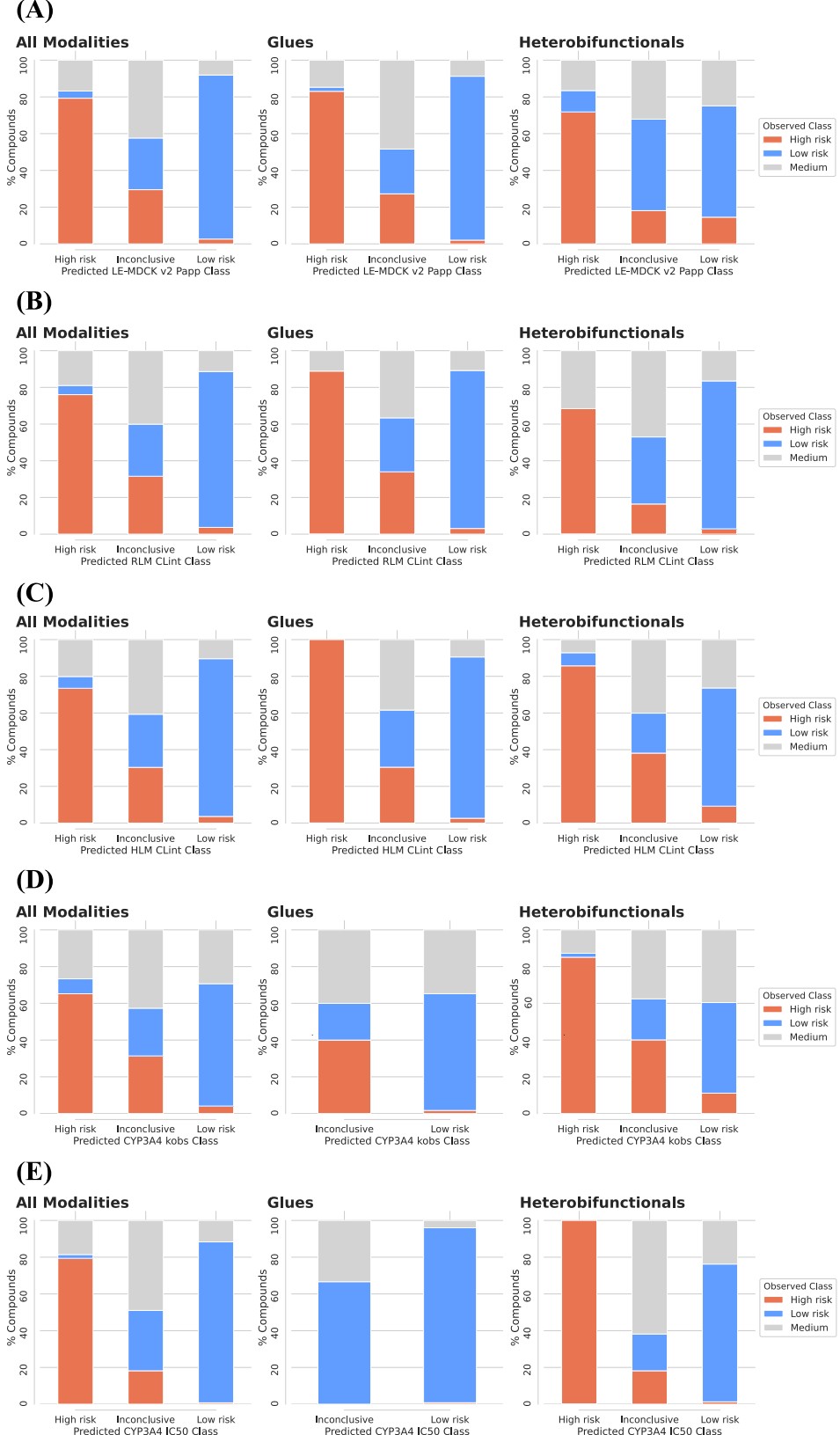

**Fig. 4 | Classification results for all modalities and targeted protein degraders (TPDs).** Reported are the percentage of compounds (y-axes) that had a given prediction (x-axes) by the global machine learning (ML) models and five properties. Prediction outputs are high risk, inconclusive (medium) or low risk categories. Results are shown for all modalities (left panel), glues (middle panel), and hetero-bifunctionals (right panel). Colors indicate the experimental three-class readout. Classification predictions are shown for passive permeability (**A**; LE-MDCK $P_{app}$),

metabolic clearance in rat liver microsomes (**B**; RLM $CL_{int}$) and human liver microsomes (**C**; HLM $CL_{int}$), CYP3A4 TDI (**D**; CYP3A4 $k_{obs}$) and reversible inhibition (**E**; CYP3A4 $IC_{50}$). The number of tested compounds were 17960 (**A**), 18322 (**B**), 18420 (**C**), 3270 (**D**) and 4377 (**E**) across all modalities; 1395 (A), 1311 (**B**), 1348 (**C**), 123 (**D**), 128 (**E**) glues; and 863 (**A**), 602 (**B**), 598 (**C**), 388 (**D**), 293 (**E**) hetero-bifunctionals. Assays are described in Table 1. Source data are provided as a Source Data file.

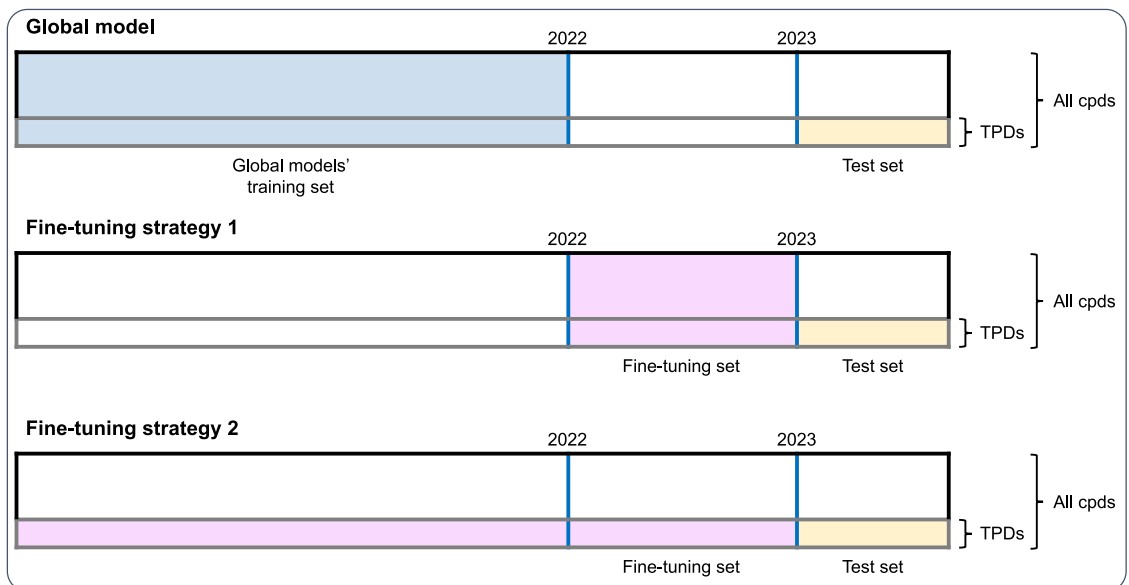

**Fig. 5 | Global model and fine-tuning strategies set-up.** Reported are the data splitting settings for global model building and fine-tuning. Global models' training set is constituted by all compounds registered in the database and measured in assays until 2021 (blue). In strategy 1, all compounds (cpds) registered and measured during 2022 were utilized for model fine-tuning (pink). In strategy 2, heterobifunctional targeted protein degraders (TPDs) that were registered and measured before 2023 were utilized for model fine-tuning (pink). In the three cases, the test set was identical and was composed by heterobifunctional TPDs registered and measured in absorption, distribution, metabolism, and excretion (ADME) assays from 1st January 2023 until 13th July 2023.

for many properties. Across all modalities, original and surrogate models had an average MAE difference of 0.04 log units, ranging from 0.01 (CYP2C6 $IC_{50}$) to 0.09 (CyLM $CL_{int}$) log units. For the prediction of glues' and heterobifunctionals' properties, MAE differences were 0.05 and 0.07 on average, respectively. Supplementary Fig. 4 reports the comparison of original and surrogate models' predictions. Interestingly, despite the presence of some outliers, the correlation between predictions of the original and surrogate MT-GNN models was consistently high across the different assays, ranging from 0.95 to 0.98 (Pearson's coefficient) and from 0.90 to 0.98 (Spearman's coefficient).

These results suggest the potential of the surrogate data sets for ML model building and applications for TPDs, and highlight which endpoints are more accurately predicted and can be useful in practice. While trained surrogate models establish a proof of principle, additional hyperparameters' optimizations and algorithms could be tested to improve QSPR models for specific properties and compounds sets. Overall, this large surrogate data set with annotated properties, including TPDs, provides new opportunities for ML-based QSPR model developments in the public domain.

## Discussion

Herein, a comprehensive evaluation of ML for the prediction of ADME and physicochemical properties of TPD molecules is firstly presented, including heterobifunctional and molecular glues submodalities. Deep learning models were generated, and prediction results showed that ADME properties such as permeability, metabolic intrinsic clearance, CYP inhibition, and lipophilicity can be successfully predicted for TPDs. Interestingly, lowest prediction errors were obtained for glues, ranging from MAE values of 0.11 (for reversible inhibition of CYP3A4) to 0.28 (for mouse metabolic clearance). Moreover, misclassification errors for high and low risk predictions were between 0 to 3.1%. For permeability, CYP3A4 inhibition, and human and rat microsomal clearance, classification errors ranged from 0 to 14.5% for heterobifunctionals. Our results suggest that predicting ADME properties for heterobifunctionals is more challenging than for glues. Transfer learning strategies were implemented to adapt the domain of ML

models and improve TPD predictions. More specifically, fine-tuning of MT-GNN models with heterobifunctionals' ADME data improved predictive performance across different ADME endpoints. The generation of a surrogate dataset based on >270,000 publicly available chemical structures, including TPDs, has also shown the potential of ML-based QSPR model building and applications for TPDs, especially glues. Predictions were highly correlated with the original in-house model predictions and were accurate for relevant endpoints such as LogD or rat metabolic clearance (RLM $CL_{int}$).

Taken together, this work indicates that ML-based QSPR models are applicable to the new modality of TPDs even though they represent a small fraction of the training set and can be further refined when additional data becomes available. With increasing TPD data availability, additional modeling strategies could be explored to further refine ADME predictions, and potentially move towards the prediction of other relevant properties from molecular structure. ML-based QSPR models are already influencing the design-make-test-analyze (DMTA) cycle in drug discovery, where only the most promising ideas are synthesized, and informative experiments are carried out. However, the use of ML models for TPDs remained marginal compared to other traditional modalities. Our findings have implications in pharmaceutical research and should increase the use of ML models for property predictions in TPD programs, potentially accelerating degraders' design with favorable ADME properties.

## Methods

### Assays' description

**LE-MDCK.** For passive permeability determination, 96-well plate permeable inserts were plated with Madin-Darby Canine Kidney (MDCK) cells and cultured for three days. The test article in dimethyl sulfoxide (DMSO) stock solution (10 mM) was added to Hanks' balanced salt solution (HBSS) to result in a final concentration of 10 μM. The HBSS buffer contained 0.02% bovine serum albumin (BSA) and 10 mM HEPES. The acceptor compartment was HBSS with 5% BSA and 10 mM HEPES. The assay was run for 120 min, determining the donor concentration at time zero and after 120 min the donor and

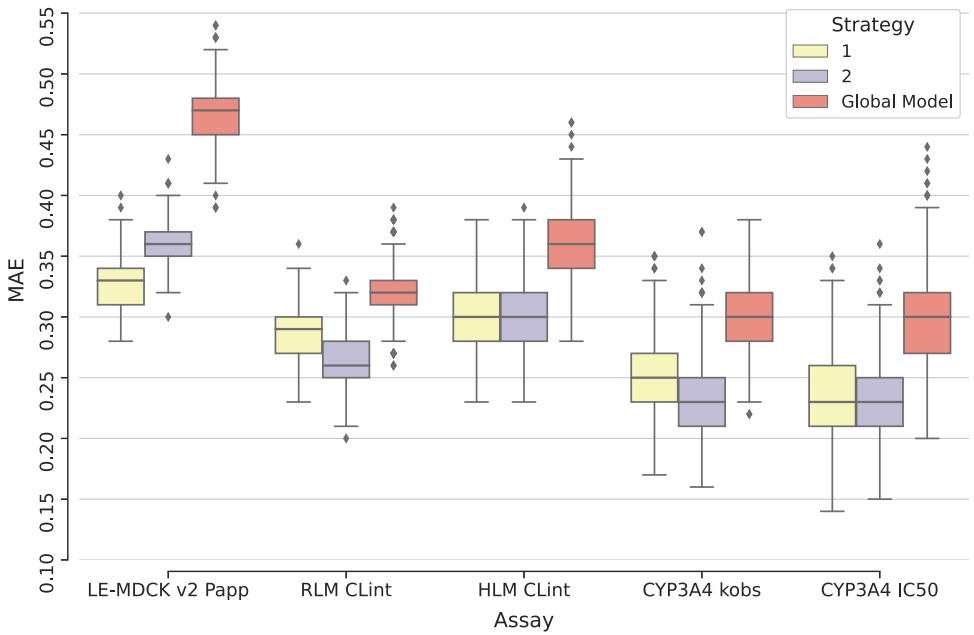

**Fig. 6 | Performance of original and refined models on heterobifunctional targeted protein degraders (TPDs).** Reported are mean absolute error (MAE) values for two fine-tuning strategies: (i) on new data (yellow) and (ii) only hetero-bifunctional data (purple), as well as the original (red) global machine learning (ML) models. Shown are bootstrapping results ($n = 1000$) for heterobifunctional TPD compounds, and five assays. Boxplots show the median (center line), and 1st and 3rd quartiles (Q1 and Q3, respectively). The error bars correspond to the Q1-(1.5*IQR) and Q3 + (1.5*IQR) range (IQR = Inter-Quartile Range). Assays are described in Table 1. Source data are provided as a Source Data file.

acceptor concentrations. The difference between version 1 (v1) and version 2 (v2) of the assays consisted in the addition of BSA[36].

**CYP metabolic stability in liver microsomes.** Microsomal incubations were performed in 384-well PCR plates at 37 °C. Test articles at a concentration of 10 mM in pure DMSO were dispensed by an acoustic dispenser to 25 μL 100 mM phosphate buffer (pH 7.4) containing 2 mM NADPH. This solution (12.5 μL, equilibrated for 10 min at 37 °C) was added to 12.5 μL liver microsomes (1 mg/mL) suspended in 100 mM phosphate buffer. At 0.5, 5, 15, and 30 min, the reactions were terminated by the addition of 10 μL acetonitrile/formic acid (93:7) containing the analytical internal standards (1 μM alprenolol and 1 μM warfarin) and transferred to a new 384-well plate containing 15 μL acetonitrile/formic acid (93:7). The stopped incubations were centrifuged at 5000 x $g$ for 15 min at 4 °C and the supernatants were analyzed by high-performance liquid chromatography–tandem mass spectrometry (LC-MS) to measure the percentage of test article remaining relative to time zero-minute incubation and determine the in vitro elimination-rate constant ($k_{mic}$). Intrinsic clearance ($CL_{int}$) was calculated by dividing $k_{mic}$ by the concentration of microsomal protein.

**PPB.** Plasma protein binding (PPB) values were mainly determined through equilibrium dialysis. Binding to proteins was measured using rapid equilibrium dialysis (RED device from ThermoFisher). Test articles were dissolved in matrix (plasma, human liver microsomes or brain homogenate). 300 μL of the matrix solutions were dispensed to the red chamber of a RED device and 500 μL 100 mM phosphate buffer to the white chamber. The RED device was sealed with a gas permeable membrane and incubated for 4 h on an orbital shaker (750 rpm) at 37 °C under 5% $CO_2$. 50 μL aliquots from both compartments were transferred to 600 μL acetonitrile containing the analytical internal standard (0.2 μM glyburide) and 50 μL buffer or matrix for a matrix match. The samples were centrifuged at 5000 x $g$ for 15 min at 4 °C and the supernatant was analyzed by LC-MS analysis for measuring test

article and internal standard. The free fraction (fu) was calculated by dividing the area ratio of the receiver compartment to the area ratio of the donor compartment. For large molecular weight compounds, PPB was measured using ultracentrifugation (UC). Test articles (5 μM) were added to 1000 μL plasma and incubated for 10 min at 37 °C in a glass vial. For the determination of the total concentration, 3 times 50 μL were added to a 96 deep-well-plate pre-filled with 600 μL acetonitrile/water (9/1) containing the analytical internal standard (0.2 μM glyburide) and 50 μL phosphate buffer. For the free fraction, an aliquot of 700 μL was centrifuged (Beckman UC Optima Max-XP) at 436,000 x $g$ for 5 h at 37 °C. At the end of the centrifugation, 3 times 50 μL of the supernatant were carefully removed and added to the 96 deep-well-plate pre-filled with acetonitrile containing the internal standard and 50 μL blank plasma for a matrix match. The 96 deep-well plate was shaked for 10 min at 300 rpm and stored over-night in a freezer at −20C° to help protein precipitation. The next day the 96 deep-well plate was centrifuged at 4500 rpm for 1 h at 4 °C. Supernatant (50 μL) was transferred in a 384 well plate with 30 μL water. The samples were analyzed by LC-MS for the measurement of test article and internal standard. High throughput dialysis (HTD) was used as a second alternative method for strong binders (% bound >99). Test articles (5 μM) were added to 700 μL plasma and incubated for 10 min at 37 °C. For the determination of the total concentration, 3 times 50 μL were added to a 96 deep-well-plate pre-filled with 600 μL acetonitrile containing the analytical internal standard (0.2 μM glyburide) and 50 μL phosphate buffer. For the free fraction, an aliquot of 100 μL was dialyzed against 100 mM phosphate buffer at pH 7.4 for 6 h in the HTD96b device (HTDialysis LLC). At the end of the incubation, 3 times 50 μL of the plasma (buffer) compartment were removed and added to the 96 deep-well-plate pre-filled with acetonitrile containing the internal standard and 50 μL blank buffer (plasma) for a matrix match. The 96 deep-well plate was shaked for 10 min at 300 rpm and stored over-night in a freezer at −20C° to help protein precipitation. The next day the 96 deep-well plate was centrifuged at 4500 rpm for 1 h at 4 °C. Supernatant (50 μL) was transferred in a 384 well plate with 30 μL

**Table 3 | Regression and classification prediction performance for refined and original ML models on heterobifunctional targeted protein degraders (TPDs)**

| Assay | Model | # Test cpds | MAE | Error low class | Error high class | % Inconclusive (Medium) predictions |
|---|---|---|---|---|---|---|
| LE-MDCK v2 $P_{app}$ | Original | 282 | 0.47 | 14% | 26.2% | 24% |
| | Retrained | | 0.34 | 2.3% | 6.9% | 34% |
| | Fine-tuned (2) | | 0.36 | 3.7% | 2.9% | 34% |
| CYP3A4 $k_{obs}$ | Original | 72 | 0.30 | 14.3% | 0% | 69% |
| | Retrained | | 0.24 | 0% | 4% | 51% |
| | Fine-tuned (2) | | 0.23 | 0% | 3.7% | 49% |
| CYP3A4 $IC_{50}$ | Original | 55 | 0.30 | 0% | 2.2% | 13% |
| | Retrained | | 0.22 | 0% | 0% | 31% |
| | Fine-tuned (2) | | 0.23 | 0% | 0% | 36% |
| RLM $CL_{int}$ | Original | 145 | 0.32 | 6.1% | 0% | 18% |
| | Retrained | | 0.28 | 4.6% | 0% | 22% |
| | Fine-tuned (2) | | 0.26 | 4% | 0% | 28% |
| HLM $CL_{int}$ | Original | 147 | 0.36 | 8.1% | 0% | 36% |
| | Retrained | | 0.31 | 3% | 14.3% | 41% |
| | Fine-tuned (2) | | 0.30 | 1.8% | 12.9% | 40% |

Reported are the mean absolute errors (MAE) for regression predictions, and percentage of misclassification errors and inconclusive (medium) predictions for categorical predictions. Results are shown for the original global models, the models after retraining, and the fine-tuned models with strategy 2. Statistics are reported for five absorption, distribution, metabolism, and excretion (ADME) assays (LE-MDCK v2 $P_{app}$, CYP3A4 $k_{obs}$, CYP3A4 $IC_{50}$, RLM $CL_{int}$, HLM $CL_{int}$), and the number (#) of test compounds (cpds) is reported. Assays are described in Table 1.

water. The samples were analyzed by LC-MS for the measurement of test article and internal standard. A calibration curve was used to define the LLOQ.

Most of the utilized data comes from RED devices but, when available, HTD or UC data was utilized instead (i.e., some >99% qualifiers were replaced). Specifically, 3–6% and 1-3% of the data was generated with HTD and UC, respectively.

**Lipophilicity.** The 1-octanol/water partitioning coefficient (LogP) was determined using a miniaturized Shake-Flask equilibrium method. Prior to start the experiment the two phases were pre-saturated, so "water-saturated 1-octanol" and "1-octanol-saturated water" were used. Samples were initially dissolved in DMSO as a 10 mM stock concentration. The samples and an internal standard were dispensed in a 1 ml deepwell plate and DMSO is evaporated prior to be dissolved in 1-octanol at a target concentration of 150 µM by shaking at 1000 rpm for 8 h. The pH 7.4 buffer was added with a phase ratio K of 1 (where K = $V_{water}/V_{octanol}$) and then the samples were shaken four hours on a shaker at 1000 rpm. The deepwell plate was centrifuged at 3000 rpm prior to phase separation. A x10 dilution for the aqueous phase and a x1000 dilution for the octanol phase are prepared and quantified by LC-HRMS against an internal standard (Dexamethasone) with a known logD = 1.9 with the following equation:

$$\log D = \log\left(\frac{\text{Analyte peak area in octanol} * 1000 / \frac{\text{IS peak area in octanol}}{0.794}}{\text{Analyte peak area in aqueous} * 10 / \text{IS peak area in aqueous}}\right)$$

This protocol was adapted from Low et al.[37].

## CYP inhibition
**CYP3A4 time-dependent inhibition (TDI).** The TDI assay was utilized to determine the first order inactivation rate ($k_{obs}$) values. Test articles were dispensed to 96-well plates, and human liver microsomes supplemented with NADPH were added to initiate the pre-incubation.

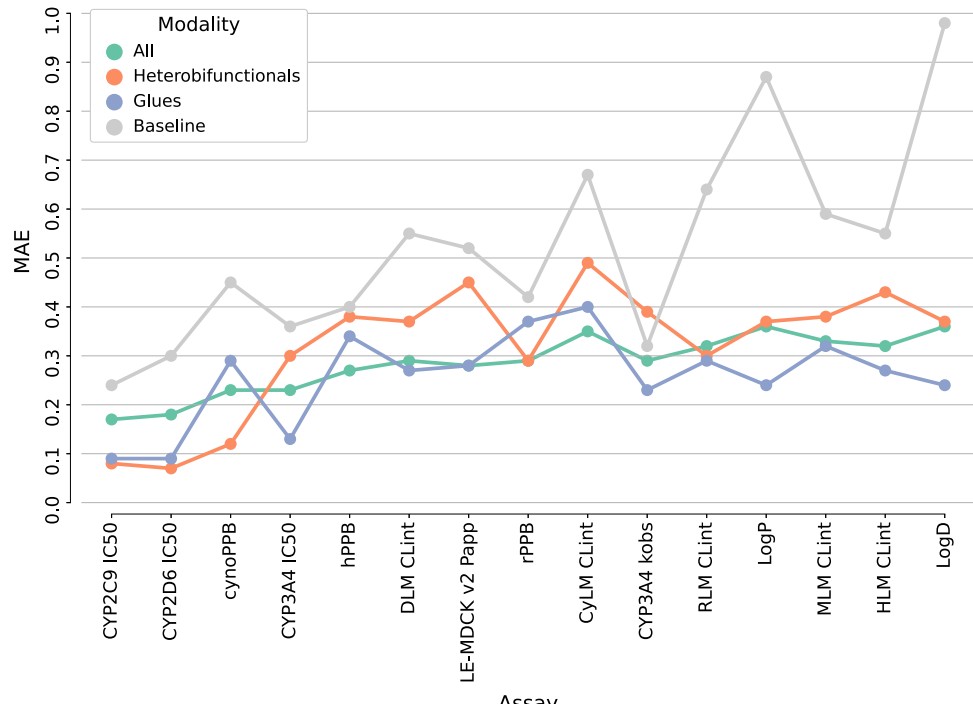

**Fig. 7 | Public global models' performance on targeted protein degraders (TPDs) and other modalities.** Public surrogate global machine learning (ML) model results are shown for fifteen assays. Reported are the mean absolute error (MAE) values for glues (blue), heterobifunctionals (orange) and all the other compounds. Assays are described in Table 1. Source data are provided as a Source Data file.

Residual CYP3A4 activity was determined after 0, 7, 16 and 32 min by the addition of midazolam (including d4-1-hydroxy-midazolam as internal standard) and incubated for six additional minutes before adding acetonitrile. Supernatants were analyzed for the CYP3A4 selective metabolite 1-hydroxymidazolam and d4-1-hydroxymidazolam using LC-MS. TDI CYP3A enzyme activity was calculated using normalized area ratios of 1-hydroxymidazolam to internal standard and plotted over the pre-incubation time. A one parameter fit using a range of 80% and a background of 20% was utilized to determine $k_{obs}$. The percentage of reversible inhibition was calculated by the area ratio at 0 min (pre-incubation) in relation to the area ratio of the control with DMSO only. In cases of strong reversible inhibition (> 50%), $k_{obs}$ values were not calculated.

**Reversible inhibition of CYP3A4, CYP2C9, and CYP2D6.** Formation rate of enzyme-specific metabolites from midazolam (CYP3A4), diclofenac (CYP2C9) and bufuralol (CYP2D6) in human liver microsomes was utilized. Substrates, internals standards and test compounds were dispensed by acoustic dispensing to a 384-well microplate. Human liver microsomes supplemented with NADPH were added to start the incubation. Plates were immediately transferred to an incubator. Incubations were stopped by the addition of acetonitrile/formic acid (93:7) and the supernatant was analyzed by LC-MS for the enzyme-specific metabolites and internal standards. The area ratios of test compounds were normalized to the average area ratio of DMSO (100% activity) and an inhibitor cocktail (0% activity) to determine the $IC_{50}$ (test compound concentration causing an inhibition of 50%) using a dose-response-model with a two-parameter fit in which 100% and 0% activity were constrained.

**Data quality.** To deliver the best possible data quality, different processes and acceptance criteria were considered. For instance, in LE-MDCK and PPB assays, data were rejected if the recovery rate was too low (< 60%). Moreover, to minimize unspecific binding issues, enzymatic incubations are done using one-well-per-data. Therefore, it is expected to extract the (bound) compound completely once the incubation is stopped (high content of organic solvent) in contrast to serial sampling approaches. This does not regard the free fraction in the incubation but increases the probability to get consistent data. The LE-MDCK assay protocol was adapted for outside rule-of-five molecules[36]. In $CL_{int}$ and CYP inhibition assays, non-specific binding is less problematic since protein is present in the incubation medium. Moreover, $F_{u,mic}$ is measured to correct $CL_{int}$ for microsomal binding. For CYP inhibition, the presence of protein decreases the non-specific binding to labware. If non-specific binding interferes too much with the assay, the data does not fit the model and no $IC_{50}$ is reported.

**Data sets for modeling**
ADME data from twenty-five assays were extracted from Novartis database and pre-processed, including apparent permeability ($P_{app}$) from two versions of the low-efflux Madin-Darby canine kidney cell line (MDCK) permeability assay, parallel artificial membrane permeability assay (PAMPA), Caco-2 permeability assay, efflux ratio from MDCK-multidrug resistance protein 1 (MDCK-MDR1) permeability assay, intrinsic clearance ($CL_{int}$) from CYP metabolic stability in liver microsomes assays for rat, human, mouse, dog, cynomolgus monkey, and minipig, plasma protein binding (PPB) for rat, human, mouse, dog and cynomolgus monkey, human serum albumin (HSA) binding, microsomal binding, brain binding, octanol-water partition (LogP) and distribution coefficients (LogP), time-dependent inhibition (TDI) of CYP3A4 (inactivation rate, $k_{obs}$), and reversible inhibition of CYP3A4, CYP2C9, and CYP2D6 (half-inhibitory constant, $IC_{50}$). Experimental data were aggregated (geometric mean) when multiple measurements were available for the same compound and assay's endpoint. Moreover, values outside the dynamic range of the assays were excluded,

and qualified values ('<'/'>') were discarded for permeability, PPB, LogP, and LogD. PPB values were transformed to fraction unbound ($F_u$) and $IC_{50}$ values from CYP reversible inhibition assays were converted to $pIC_{50}$ with a negative logarithmic transformation. Logarithmic transformations were applied to the rest of the assay endpoints, except to LogP and LogD. All above-mentioned assays were utilized for model training, but only a fraction of them were used for model evaluation due to data availability. For instance, some assays are requested less often (e.g. monkey $CL_{int}$ compared to rat $CL_{int}$) or were deprecated in favor of newer version (e.g. LE-MDCK version 2 assay) or other technologies (e.g. Caco-2 was deprecated). Data set statistics are discussed and reported below.

**Global QSPR models' description**
Four multi-task graph neural networks (MT-GNN) global models were generated and evaluated herein: *Permeability* (5-task model), *Clearance* (6-task model), *Binding/Lipophilicity* (10-task model), and *CYP inhibition* (4-task model). The models were ensembles of a message-passing neural network (MPNN) followed by a feed-forward deep neural network (DNN)[25,26]. These DNNs facilitate MT through the consideration of multiple output neurons (one per task). To enable MT model training with sparse labels, a masked loss function was utilized, and missing values were not considered for backpropagation[22]. Previous investigations indicated that especially for sparse experimental data, MT learning can provide an advantage compared to single-task models[38]. For all models, rectified linear unit (ReLU) was the activation function, a batch size of 50 was considered, and the models were trained with the consideration of early stopping (with a validation set of 10% with scaffold split). Learning rate was varied from an initial value of 0.0001 to a maximum of 0.001 linearly and decreased until 0.0001 exponentially. Mean aggregation was applied to convert atomic vectors into molecular vectors. Supplementary Table 2 reports details about each model's architecture. Models evaluated herein are available for ADME assays' predictions internally at Novartis. Prior to selecting MT-GNN as a modeling approach, a variety of molecular features, ML methods, and hyperparameters were benchmarked[2,20].

**Prediction tasks**
Table 1 reports the fifteen prediction tasks that were evaluated, including the assay, measured property, and MT-GNN model where they were included. Numerical assay thresholds were used to categorize experimental read-outs into 'high risk' and 'low risk' classes[1]. Compound optimization accounts for multiple properties, which are measured with assays of varying resolution (different experimental errors)[7]. Therefore, this assay risk categorization defined by assay experts accounts for experimental variability[20] and facilitates decisions during multiparameter optimization. Following this three-class concept, property predictions were also converted to a three-output classification ('low risk', 'medium', 'high risk') using the same thresholds utilized for experimental read-outs, which are reported in Table 1. The percentage of compounds belonging to each category is reported in Supplementary Table 1. When applying such assay thresholds to ML outputs, predictions in the 'medium' category can be disregarded. By only considering 'high risk' and 'low risk' predictions for decision-making, ML models' precision improves. As in other works[20,21,39], predictions in the medium range can also be termed 'inconclusive (medium)' predictions, which ideally are not a large fraction. This regression-based classification approach was previously proposed[20,40].

**Data splitting**
ML models were evaluated prospectively (with newly registered and measured molecules)[2]. Such evaluation scenario is commonly

referred to as temporal validation or time-split[41]. All MT-GNN global models were trained with compounds registered until the end of 2021 and evaluated with compounds registered from 1st January 2022 until 13th July 2023. Global models were evaluated on glue and heterobifunctional TPDs, and predictive performance on these two TPD modalities was compared to performance across all modalities. For the prediction tasks under evaluation, Fig. 1 reports the number of training and test set compounds per each property and modality. The *Permeability model* was trained on 206,347 compounds, which included 2,732 heterobifunctionals and 2673 glues. From those compounds, 20,041 had LE-MDCK v2 $P_{app}$ measurements, including 1,608 heterobifunctionals and 1404 glues. *Clearance*, *Binding/Lipophilicity*, and *CYP inhibition* models were generated with 223,025, 92,464, and 65,701 compounds, respectively. Supplementary Table 3 reports the number of training compounds for all the tasks included in global models.

## Performance metrics

For regression models, performance was estimated with the mean absolute error (MAE).

$$\text{MAE} = \frac{1}{n}\sum_{i=1}^{n}||y_i - \hat{y}_i|| \tag{1}$$

where $y$: experimental value, $\hat{y}$: prediction, and n: number of compounds.

Classification models were evaluated with the average precision on low and high classes. Precision is the percentage of compounds that were predicted 'high' ('low') and indeed had a 'high' ('low') property value.

$$\text{Precision} = \frac{\text{TP}}{\text{TP} + \text{FP}} \tag{2}$$

where TP: true positives, and FP: false positives.

Misclassification or error rates were also calculated, as the percentage of compounds that were predicted to have a 'high' ('low') property value and had a 'low' ('high') measured value. Finally, the percentage of inconclusive (medium) predictions is computed as the fraction of molecules with a predicted property value in the 'medium' range. For those compounds, no 'high' or 'low' property prediction is given.

## Models' refinement

Some prediction tasks showing margin for improvement were identified and, when data availability allowed, model refinement was carried out. Specifically, three MT-GNN models were optimized with new data attempting at improving LE-MDCK $P_{app}$ (permeability model); RLM $CL_{int}$ and HLM $CL_{int}$ (clearance model); and CYP3A4 $k_{obs}$ and CYP3A4 $IC_{50}$ (CYP model) predictions.

Transfer learning was adopted with the purpose of optimizing model parameters for heterobifunctional TPD compounds. Instead of focusing on transferring knowledge to previously unseen tasks, which is more common in the field[42–44], transfer learning was applied under the paradigm of domain adaptation[29,45]. The transfer learning approach utilized was model fine-tuning with weights initialization, where the weights of the original model (pre-trained model) were adjusted with new data[28,46]. Two fine-tuning strategies were investigated and are illustrated in Fig. 5.

**Strategy 1** (New data). Global ML models were fine-tuned with all new data generated during the previous year. Here, the model was fine-tuned with compounds registered and measured in 2022.

**Strategy 2** (TPD-specific data). The original global ML model was fine-tuned with heterobifunctional TPDs' data (registered and measured before 2023).

Both strategies were evaluated on 328 heterobifunctional TPDs synthesized and measured in 2023. Supplementary Table 4 reports the number of compounds in the training, fine-tuning, and test sets.

## Public structures and surrogate data set

A data set of public structures was gathered from ChEMBL[33], ZINC[34], and PROTAC-DB[35]. ChEMBL and ZINC data sets were prepared according to Rodriguez-Perez and Bajorath[47]. Structures from the three data sources were standardized, including hydrogen bonds removal, metal disconnection, molecule normalization, reionization, and stereochemistry assignation, with the RDKit module *rdMolStandardize*, and canonicalized[48]. After such pre-processing, the data set was composed by 273,706 molecules from ChEMBL (70,465), ZINC (199,972) and PROTAC-DB (3,269). To get physicochemical and ADME property values for this large set of molecules, the internal Novartis ML models were utilized. Twenty-five compound properties were predicted for each of the molecules to generate a surrogate data set. This surrogate data set is provided as Supplementary Data.

## Reporting summary

Further information on research design is available in the Nature Portfolio Reporting Summary linked to this article.

## Data availability

Source data are provided with this paper. The data used to generate some models in this study (*novartis_tpd_NatureCommunications2024*) are proprietary to Novartis. These data are not publicly available due to intellectual property restrictions. The surrogate data set with publicly available structures (including TPDs) and predicted properties is given as Supplementary Data. Source data are provided with this paper.

## Code availability

Supplementary Software is provided with this publication to build the four MT-GNN global models and obtain property predictions.

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

## Acknowledgements

The authors thank Francis J. Prael III and Thomas Zoller for their help in dataset preparation, Paolo Tosco for discussions about molecular similarity, and Markus Trunzer, Bernard Faller, Gaëlle Chenal, and Stephane Rodde for assistance with the assays' descriptions. Thanks to all former and current Novartis colleagues who helped generating the data used for modeling. M.T.D.H. thanks the Translational Medicine Data Science Academy program at Novartis Biomedical Research.

## Author contributions

G.G. and R.R.P. conceived and supervised the study; R.R.P. and G.P. built the internal models; G.P. evaluated the internal models; M.T.D.H. built and evaluated the public models; G.P., M.T.D.H., and R.R.P. prepared the figures and analyzed the results; R.R.P. and G.P. wrote the manuscript; all authors discussed the results and revised the manuscript.

## Competing interests

The authors declare no competing interests.
