## [Peer Review File · Nature Communications]

Application of machine learning models for property prediction to targeted protein degradersREVIEWER COMMENTS

Reviewer #1 (Remarks to the Author):

Key Results

This report is the first comprehensive evaluation of ML for the prediction of ADME and physiochemical properties of TPD molecules, including heterobifunctional and molecular glue submodalities. In a recent pharmaceutical industry perspective paper, Volak et al (2023) DMD indicated there was a knowledge gap in how TPDs performed in in silico and machine learning models and this paper addresses that gap. The findings here complement that paper by showing that TPDs are no different than other small molecule modalities.

Validity and Robustness of Data Interpretation and Conclusions

The authors sometimes make broad statements which are not correct based on the data presented in tables and figures. For example, the last paragraph of page 21 states that "Moreover, misclassification errors were at most 14.5% and for glues they were always lower than 3%," however, RLM CLint has an error of 3.1%. This can easily be corrected. Another example is in the same paragraph which has "the percentage of inconclusive predictions was consistently lower than 35%," but there are errors of 38% and 51% in some assays. Adding the word generally will correct this. At the top of page 27 there is a statement of "fine-tuning with bifunctional TPD data always yielded the lowest misclassification errors," but based on the data fine-tuning did not "always" yield the lowest errors (see low class for MDCK).

Significance of Conclusions

Machine learning is an evolving field, and this manuscript shows that model predictions for TPDs is no better than any other small molecule modality. However, in my opinion, predictions still are not good with most assays having >20% of compounds not able to be predicted and one assay having >50% of compounds without predictions even when using quite broad (high/low risk) classification buckets. These models can be used to prioritize chemistry synthesis but it is a broad statement in the conclusions to state that these models may be able to predict PK from structure. We are a long way from those type of predictions.

Data and Methodology (quality of data, of presentation, Note areas out of reviewer's expertise)

Suggest revision if paper is found to have sufficient significance for publication in Nature Communications.

The authors do not provide any context on the definition of "other modalities." Does the other modalities category only include small molecules and if so, within that small molecule set, what percentage would be considered beyond the Lipinski's Rule of 5 (bRo5)? The addition of a summary of the additional properties of the different chemical categories, beyond MW, would be useful. I notice that some of the compounds in other modalities have MW >1000. Are these peptides as these are not a fair comparison with bifunctional degraders or glues?

Related to the definition of other modalities, there is little information about the diversity of heterobifunctional or glue degraders. If most of the glue degraders are Cereblon degraders, then the predictions are going to be better than other modalities, mostly because there is little structural diversity. The limited range of ADME activities of glues can be observed for some assays in Figure S1. It would be helpful to provide some sort of benchmarking of the structural diversity of the categories beyond that provided in Figure 3.

The manuscript has no description of the quality of data used in the models. Heterobifunctional degraders are prone to non-specific binding in many ADME assays which can impact data quality by lowering recovery. However, glues tend not to have these issues and data would be of higher quality unless only the highest quality data (or at least comparable to glues) for heterobifunctional degraders was used.

The ADME methods section in Supplemental needs to have key assay details added. There are no methods for HTD and UC PPB and many of the metabolic assays, which are sensitive to solvent concentrations, do not specify the DMSO and/or acetonitrile concentrations.

Reviewer suggests changing the Inconclusives category to Inconclusive (Medium) Predictions as Inconclusives is not a word and slang.

The manuscript starts using heterobifunctional degraders then starts calling them bifunctionals. Suggest changing all to heterobifunctional degraders.

Table 1 lists MDCK Papp of >5 as high risk but this is low risk.

LM abbreviations are not standard. Usually hLM is HLM and rLM is RLM.

Log P and LogD descriptions in Table 1 are switched.

Table 3 MDCK cell needs to be corrected by expanding/merging the cell to include the fine-tuned model.

Suggest the authors re-read the manuscript for typos and incorrect tense.

Reviewer experience is in DMPK, ADME, and TPDs and cannot comment on the validity of the message-passing neural network (MPNN) coupled with a feed-forward deep neural network (DNN) machine-learning approaches used in the manuscript to generate the comparisons.

Analytical Approach

To the best of my knowledge, the approach seems valid, but ML and related statistics are not my expertise.

Suggested Improvements (experiments or data)

One experiment which would be beneficial to the reader is to compare the prediction of ADME properties of heterobifunctional degraders to bRo5 compounds in the "other modalities" category. If data used for ML predictions was not curated for high recovery and high quality results, then it might be useful to re-test the models with a high quality data set for TPDs.

Clarity and Context

The manuscript is easy to read even with the complicated methodology. There are some minor edits that are needed to improve readability.

References

References are appropriate. It may be beneficial and provide more evidence for the need for this work by referencing Volak et al (2023) DMD.

Constructive Feedback

As mentioned in other areas, the authors should carefully review their manuscript to ensure that there are not blanket statements which do not match the results presented in tables and figures. For example, do not state always when there is an data set showing that the finding is not always observed. Also, make sure that the ranges and numbers in the text are correct based on the data.

Reviewer #2 (Remarks to the Author):

In this computational study, Peteani et al. investigate the suitability of conventional ML-based QSPR models for ADME predictions when applied to targeted protein degraders. The work is undoubtedly timely and addresses a significant open question in the field. The abstract clearly states the findings: ML-QSPR can be effectively used for molecular glue degraders and bifunctional degraders, such as PROTACs, with reasonable accuracy. As anticipated, ADME prediction for bifunctional (larger) molecules presents greater challenges due to the chemical space covered by the training sets. In my view, these conclusions are relevant and promising, especially as the paper demonstrates that "global" ADME models are effective in this context.

While I generally support this work, I have some comments:

1. The study relies on data from the Novartis database, specifically modeling 25 assays using ML-QSPR, which encompass cytochromes, permeability, plasma protein binding, microsomal binding, and more. These datasets are invaluable, and I recognize that sharing the data might be restricted due to IP concerns. However, the authors might contemplate whether they could, at the very least, share or make their models available online for the broader community, possibly in an encrypted manner. I notably missed a "code and data availability statement" in the manuscript. Including such a statement could greatly amplify the work's impact, assuming IP constraints can be addressed.

2. The modeling strategy is sound, employing an ensemble of MPNN and six multi-task (MT) global models. I would appreciate more details in the methods section, specifically:

Within each MT model, how were missing values addressed? Based on Fig 1, the number of compounds varies for each individual task, indicating the presence of missing values. How extensive was the hyperparameter tuning?

3. I appreciate the "three-class concept" (low risk, medium/inconclusive, high risk) applied to regression tasks. In Table 1, it would be beneficial to know the percentage of molecules categorized under the high-risk and low-risk regions.

4. It's commendable that evaluations were conducted prospectively (time-split). However, the manuscript doesn't clarify how these models are subsequently utilized (if at all) within Novartis. Insights into the 'real-world' significance of these models would be valuable, perhaps with a quantitative analysis of reduced preclinical attrition rates in TPD programs within the company's pipelines.

5. The authors offer a compelling demonstration of transfer learning, showing that the inclusion of new data points enhances prediction accuracy. The added data covers a span of one year. It would be insightful to investigate more immediate applications of transfer learning, such as (a) on a series/scaffold-specific basis (how many molecules from a specific series/scaffold are needed to boost accuracy for that particular series?) or (b) by incorporating molecules/assays from public sources. Additionally, it would be good to determine the impact of transfer learning when a random subset of molecules is used, in contrast to the 2022 subset.

6. Interestingly, model retraining doesn't appear to be successful. Can the authors provide an explanation for this?

7. Overall, there isn't much ML-QSPR experimentation (no new architectures or descriptors were developed). As such, greater emphasis should be placed on the analysis of predictions. In this regard, I believe the authors could delve deeper into the attributes of incorrectly predicted molecules, or even the inconclusive ones. Do these molecules possess specific characteristics that make them harder to predict?

8. Regrettably, while the results are interesting, the quality and visual allure of the figures don't meet expectations. There are inconsistencies in font sizes and types, varying styles in the axes (e.g., ticks vs. non-ticks, unlabeled axes), and issues with readability (for instance, Figure 1 has minuscule fonts). Moreover, the panels in Figure 5 should be consolidated into a single figure

block.

9. The UMAP figure is insufficiently explained and appears pixelated in the PDF version. Which descriptors or fingerprints were utilized?

10. In line 291, "Figure 3a" should be corrected to "Figure 4a."

On the whole, this research conveys an important and positive message to the TPD community. However, not all modelers, particularly those in academia, will have the opportunity to work with such rich and extensive datasets. Therefore, as mentioned earlier, it would be good if the authors made an effort to share the computational resources presented here with the wider community in some form.

Reviewer #3 (Remarks to the Author):

Thank you for allowing me to review the manuscript by Peteani et al., "Are machine learning-based QSPR models applicable to targeted protein degraders?" (NCOMMS-23-44731).

Comments:

The authors constructed deep learning-based prediction models to predict the important ADME properties focused on glues and heterobifunctional TPDs. ADME properties prediction for TPDs succeeded, and more effort was put into transfer learning. The authors also showed the difficulty of predicting bi-functionals. It would be informative and contribute to future PK studies to examine the various methods to predict ADME properties with large-scale data. Their objectives and results have been well demonstrated. Although it is recommended to add some additional analyses to prove the conclusion more clearly, I recommend publication after the minor revisions noted below.

Recommendation:

Minor comment:

1. Regrettably, analyses using in-house company data are generally not fully disclosed in the data set. I would be glad if you could explain in as much detail as possible the distribution of the data set and their chemical space. Please visualize the difference in chemical space and molecular weight distribution between the training set and the test set in each compound group, such as all, bfx, and glues by UMAP. The degree of similarity between past and recent data is essential information that directly affects accuracy.
2. The data extracted from the 25 assays differ in number for each parameter. How did you treat the data for which no numerical values were present in the dataset for these multi-tasking studies? Please add an explanation in the Materials and Methods section. Also, have you considered how the proportion of data for which no numerical values are present affects the results?
3. Please reconsider Figure 2; the difference between Strategy 1 and 2 cannot be taken from Figure 2.
4. p18, section 3.2, Please describe in more detail how you constructed the baseline predictor.

Point-by-point response

Our responses and comments are in red.

REVIEWER COMMENTS

Reviewer #1 (Remarks to the Author):

Key Results

This report is the first comprehensive evaluation of ML for the prediction of ADME and physiochemical properties of TPD molecules, including heterobifunctional and molecular glue submodalities. In a recent pharmaceutical industry perspective paper, Volak et al (2023) DMD indicated there was a knowledge gap in how TPDs performed in in silico and machine learning models and this paper addresses that gap. The findings here complement that paper by showing that TPDs are no different than other small molecule modalities.

Response: We appreciate reviewer's comments about the relevance of our study. We have included the reviewer's comment in the Abstract and Conclusions to further motivate our work.

Validity and Robustness of Data Interpretation and Conclusions

The authors sometimes make broad statements which are not correct based on the data presented in tables and figures. For example, the last paragraph of page 21 states that "Moreover, misclassification errors were at most 14.5% and for glues they were always lower than 3%," however, RLM CLint has an error of 3.1%. This can easily be corrected. Another example is in the same paragraph which has "the percentage of inconclusive predictions was consistently lower than 35%," but there are errors of 38% and 51% in some assays. Adding the word generally will correct this. At the top of page 27 there is a statement of "fine-tuning with bifunctional TPD data always yielded the lowest misclassification errors," but based on the data fine-tuning did not "always" yield the lowest errors (see low class for MDCK).

Response: Thanks for pointing that out. We have now corrected these statements.

Significance of Conclusions

Machine learning is an evolving field, and this manuscript shows that model predictions for TPDs is no better than any other small molecule modality. However, in my opinion, predictions still are not good with most assays having >20% of compounds not able to be predicted and one assay having >50% of compounds without predictions even when using quite broad (high/low risk) classification buckets. These models can be used to prioritize chemistry synthesis but it is a broad statement in the conclusions to state that these models may be able to predict PK from structure. We are a long way from those type of predictions.

Response: We have modified the statement regarding the possibility of predicting PK from structure, which is based in our experience and literature, but it is not supported by data shown in this work. Instead of 'potentially move towards the prediction of in vivo PK from molecular structure' we have written 'potentially move towards the prediction of other relevant properties from molecular structure.'

On the other hand, the models provide very low misclassification rates (e.g. from 0 to 13% for fine-tuned models for heterobifunctionals and from 0% to 3.1% for glues). We have observed that the precision of these MT-GNN models approached the reproducibility of the experimental assay (Mol. Pharmaceutics 2023, 20, 383–394). Finally, 'inconclusive' predictions correspond to predictions in the medium range, and between 7% (CYP2C9 IC50) and 45% (hPPB) compounds also had experimental measurements in the medium category. Experimental results in the 'medium range' often change category (to high- or low-risk) if the measurement is repeated (i.e. are also in an 'inconclusive (medium)' category according to the ground truth from the assay readout. We have pointed out these details in the main text (pg.17) and reported the statistics of class distributions (including the medium class) in Table S1.

Data and Methodology (quality of data, of presentation, Note areas out of reviewer's expertise)

Suggest revision if paper is found to have sufficient significance for publication in Nature Communications.

The authors do not provide any context on the definition of “other modalities.” Does the other modalities category only include small molecules and if so, within that small molecule set, what percentage would be considered beyond the Lipinski’s Rule of 5 (bRo5)? The addition of a summary of the additional properties of the different chemical categories, beyond MW, would be useful. I notice that some of the compounds in other modalities have MW >1000. Are these peptides as these are not a fair comparison with bifunctional degraders or glues?

Response: More information is provided about the definition of “other modalities” and the percentage of compounds beyond the Ro5 (bRo5), which is 34% (pg.11). We have also reported the distribution of additional properties of the different chemical categories, including hydrogen bond acceptors (HBA), donors (HBD), topological polar surface area (TPSA), calculated LogP (cLogP), and number of rotational bonds. This additional information is provided in Figure 2a. Only a small fraction of peptides is included in the ‘other modalities’ set, which will not dominate the reported statistics.

Related to the definition of other modalities, there is little information about the diversity of heterobifunctional or glue degraders. If most of the glue degraders are Cereblon degraders, then the predictions are going to be better than other modalities, mostly because there is little structural diversity. The limited range of ADME activities of glues can be observed for some assays in Figure S1. It would be helpful to provide some sort of benchmarking of the structural diversity of the categories beyond that provided in Figure 3.

Response: We have included additional information about the diversity of heterobifunctional and glue degraders by reporting the distribution of calculated properties such as molecular weight, HBA, TPSA..., as detailed above (pg.11-12 and Figure 2a). The figure was inspired in Fig.2 from Volak et al (2023) DMD, which the reviewer recommended. Moreover, the percentage of glues and heterobifunctionals bRo5 is also reported in the main text (19% and 100%, respectively). This additional information is complemented by the UMAP chemical space representation in Figure 2b, which has been regenerated with more quality.

The manuscript has no description of the quality of data used in the models. Heterobifunctional degraders are prone to non-specific binding in many ADME assays which can impact data quality by lowering recovery. However, glues tend not to have these issues and data would be of higher quality unless only the highest quality data (or at least comparable to glues) for heterobifunctional degraders was used.

Response: We have added a section about ‘Data quality’ in Supplementary information: *To deliver the best possible data quality, different processes and acceptance criteria were considered. For instance, in LE-MDCK and PPB assays, data were rejected if the recovery rate was too low (<60%). Moreover, to minimize unspecific binding issues, enzymatic incubations are done using one-well-per-data. Therefore, it is expected to extract the (bound) compound completely once the incubation is stopped (high content of organic solvent) in contrast to serial sampling approaches. This does not regard the free fraction in the incubation but increases the probability to get consistent data. The LE-MDCK assay protocol was adapted for outside rule-of-five molecules (J Pharm Sci (2021), 110(6):2562-2569). In CL_{int} and CYP inhibition assays, non-specific binding is less problematic since protein is present in the incubation medium. Moreover, $F_{u,mic}$ is measured to correct CL_{int} for microsomal binding. For CYP inhibition, the presence of protein decreases the non-specific binding to labware. If non-specific binding interferes too much with the assay, the data does not fit the model and no IC_{50} is reported.*

The ADME methods section in Supplemental needs to have key assay details added. There are no methods for HTD and UC PPB and many of the metabolic assays, which are sensitive to solvent concentrations, do not specify the DMSO and/or acetonitrile concentrations.

Response: We have added more details for HTD PPB, UC PPB and the metabolic assays in Supplementary Information. Metabolic stability assays’ description includes information to enable the reader to calculate the DMSO concentration in the incubate. With the acoustic dispensing (ECHO) it is

possible to dispense a few nL which enable a direct dilution and the lowest possible solvent concentration when starting from 10 mM DMSO stock solutions (e.g. 2.5 nL 10 mM DMSO stock in 25 μ L incubate leads to 1 μ M compound and 0.01% DMSO in the incubation).

Reviewer suggests changing the Inconclusives category to Inconclusive (Medium) Predictions as Inconclusives is not a word and slang.

Response: The terminology has been changed.

The manuscript starts using heterobifunctional degraders then starts calling them bifunctionals. Suggest changing all to heterobifunctional degraders.

Response: We now use 'heterobifunctionals' instead of 'bifunctionals'.

Table 1 lists MDCK Papp of >5 as high risk but this is low risk.

Response: Thanks for detecting this typo, which has been now corrected.

LM abbreviations are not standard. Usually hLM is HLM and rLM is RLM.

Response: We have changed the abbreviations.

Log P and LogD descriptions in Table 1 are switched.

Response: Thanks for noticing this typo, which has been fixed.

Table 3 MDCK cell needs to be corrected by expanding/merging the cell to include the fine-tuned model.

Response: The cell has been merged.

Suggest the authors re-read the manuscript for typos and incorrect tense.

Response: The suggestion has been followed.

Reviewer experience is in DMPK, ADME, and TPDs and cannot comment on the validity of the message-passing neural network (MPNN) coupled with a feed-forward deep neural network (DNN) machine-learning approaches used in the manuscript to generate the comparisons.

Analytical Approach

To the best of my knowledge, the approach seems valid, but ML and related statistics are not my expertise.

Suggested Improvements (experiments or data)

One experiment which would be beneficial to the reader is to compare the prediction of ADME properties of heterobifunctional degraders to bRo5 compounds in the "other modalities" category. If data used for ML predictions was not curated for high recovery and high quality results, then it might be useful to re-test the models with a high quality data set for TPDs.

Response: Thanks for the suggested improvements, which have been included and explained above.

Clarity and Context

The manuscript is easy to read even with the complicated methodology. There are some minor edits that are needed to improve readability.

Response: Thanks a lot, we have done some modifications attempting at further improving readability.

References

References are appropriate. It may be beneficial and provide more evidence for the need for this work by referencing Volak et al (2023) DMD.

Response: The reference has been included in the Introduction.

Constructive Feedback

As mentioned in other areas, the authors should carefully review their manuscript to ensure that there are not blanket statements which do not match the results presented in tables and figures. For example, do not state always when there is an data set showing that the finding is not always observed. Also, make sure that the ranges and numbers in the text are correct based on the data.

Response: Thanks for pointing this out. These statements are now fixed.

Reviewer #2 (Remarks to the Author):

In this computational study, Peteani et al. investigate the suitability of conventional ML-based QSPR models for ADME predictions when applied to targeted protein degraders. The work is undoubtedly timely and addresses a significant open question in the field. The abstract clearly states the findings: ML-QSPR can be effectively used for molecular glue degraders and bifunctional degraders, such as PROTACs, with reasonable accuracy. As anticipated, ADME prediction for bifunctional (larger) molecules presents greater challenges due to the chemical space covered by the training sets. In my view, these conclusions are relevant and promising, especially as the paper demonstrates that "global" ADME models are effective in this context.

Response: We appreciate the positive comments from the reviewer.

While I generally support this work, I have some comments:

1. The study relies on data from the Novartis database, specifically modeling 25 assays using ML-QSPR, which encompass cytochromes, permeability, plasma protein binding, microsomal binding, and more. These datasets are invaluable, and I recognize that sharing the data might be restricted due to IP concerns. However, the authors might contemplate whether they could, at the very least, share or make their models available online for the broader community, possibly in an encrypted manner. I notably missed a "code and data availability statement" in the manuscript. Including such a statement could greatly amplify the work's impact, assuming IP constraints can be addressed.

Response: We agree that the datasets are invaluable and, as the reviewer indicated, data accessibility is restricted. Sharing the models' based on such data is also prohibitive due to the possibility of reverse engineering. We believe that the conclusions of the paper and learnings from our investigations would be of relevance for the scientific community even in the absence of the data availability. Specifically, our work demonstrates the validity of ML-based QSPR models for TPD property prediction, gives access to state-of-the-art models' performance on this new modality, and show approaches to further improve predictions. We anticipate that our results will also assist in putting performance values in context with some realistic industrial prospective set-ups, and thus, help other researchers to apply and further develop appropriate modeling strategies for TPDs. We have highlighted this in the Discussion (pg.27).

However, we have also worked towards improving the reproducibility of our work and share resources that could further help in the development of new ML systems for TPDs. For that, we generated a public surrogate data set with >270,000 data points, following an approach proposed by Tetko et al. JCAMD 2005. The data is shared as Supplementary Data. New 'public surrogate models' were trained with compound structures from ChEMBL, ZINC, and PROTAC-DB databases and annotations from our internal model predictions. Surrogate public models were validated and discussed in the revised manuscript in pg.25-26 (*Results. Public surrogate data and ML model*) as well as pg.33 (*Methods. Public structures and surrogate data set*). Code to generate the models and get predictions is also provided as Supplementary Software. Code and data availability statements has been included in the revised manuscript.

2. The modeling strategy is sound, employing an ensemble of MPNN and six multi-task (MT) global models. I would appreciate more details in the methods section, specifically: Within each MT model, how were missing values addressed? Based on Fig 1, the number of compounds varies for each individual task, indicating the presence of missing values. How extensive was the hyperparameter tuning?

Response: We have now added information about how the missing values were considered (pg.30). Specifically, we have utilized a custom loss function that masks missing values and can consider only available labels for model training (backpropagation). ML models evaluated herein are available for ADME assays' predictions internally at Novartis, and a variety of molecular representations and ML algorithms and hyperparameters were compared (Mol. Pharmaceutics 2023, 20, 3, 1758–1767). For the Clearance model, an exhaustive description of the model selection and design is also available in Mol. Pharmaceutics 2023, 20, 383–394. Additional information about the models has been included in pg.30.

3. I appreciate the "three-class concept" (low risk, medium/inconclusive, high risk) applied to regression tasks. In Table 1, it would be beneficial to know the percentage of molecules categorized under the high-risk and low-risk regions.

Response: Table S1 has now been added that reports the percentage of molecules under high-risk and low-risk regions, and it is mentioned in the main text (pg.17 and 31).

4. It's commendable that evaluations were conducted prospectively (time-split). However, the manuscript doesn't clarify how these models are subsequently utilized (if at all) within Novartis. Insights into the 'real-world' significance of these models would be valuable, perhaps with a quantitative analysis of reduced preclinical attrition rates in TPD programs within the company's pipelines.

Response: We have now included a discussion about the practical application of these models at Novartis and the 'real-world' significance in the Conclusions section: *ML-based QSPR models are already influencing the design-make-test-analyze (DMTA) cycle in drug discovery, where only the most promising ideas are synthesized and informative experiments are carried out. However, the use of ML models for TPDs remained marginal compared to other traditional modalities. Our findings have implications in pharmaceutical research and should increase the use of ML models for property predictions in TPD programs, potentially accelerating degraders' design with favorable ADME properties.*

However, a quantitative analysis of reduced preclinical attrition rates is prohibitive in this case. TPD programs have started using the predictions of ML models very recently, in light of the results of this investigation, and we would need more time to assess the long-term impact.

5. The authors offer a compelling demonstration of transfer learning, showing that the inclusion of new data points enhances prediction accuracy. The added data covers a span of one year. It would be insightful to investigate more immediate applications of transfer learning, such as (a) on a series/scaffold-specific basis (how many molecules from a specific series/scaffold are needed to boost accuracy for that particular series?) or (b) by incorporating molecules/assays from public sources. Additionally, it would be good to determine the impact of transfer learning when a random subset of molecules is used, in contrast to the 2022 subset.

Response: These are really interesting ideas and we have studied some of them. For instance, we have investigated immediate applications of transfer learning for specific drug discovery projects, working on a confined chemical space (specific series). In such context, we have assessed the influence of the number of compounds utilized for the transfer learning on model performance. Such investigations were quite extensive and did not exclusively focus on TPDs (to have larger data sets). We believe that including those analyses in the current manuscript would attenuate the message and we have recently submitted a manuscript entitled 'Adapting deep learning QSPR models to specific drug discovery projects', which is more focused on the methodologies and benchmarks of different transfer learning strategies (and not only about TPDs).

6. Interestingly, model retraining doesn't appear to be successful. Can the authors provide an explanation for this?

Response: Model retraining improves predictive performance, so we would say that it is also a successful strategy. However, transfer learning can give more accurate predictions, which suggests that using a pre-trained global model and subsequently refine predictions using a relevant data set (e.g. new modality) might be a more successful strategy. This has been further discussed in pg.23 of the revised manuscript.

7. Overall, there isn't much ML-QSPR experimentation (no new architectures or descriptors were developed). As such, greater emphasis should be placed on the analysis of predictions. In this regard, I believe the authors could delve deeper into the attributes of incorrectly predicted molecules, or even the inconclusive ones. Do these molecules possess specific characteristics that make them harder to predict?

Response: ML-QSPR experimentation has focused on models' evaluation and transfer learning for TPDs. Following the reviewer's suggestion, we attempted to find attributes of incorrectly predicted molecules. In such error analysis, the distribution of six calculated molecular descriptors (from Lipinski's Rule-of-5) were compared for correctly classified and misclassified molecules. Please, find this figure in Supplementary Information for Review. Overall, correctly classified and misclassified molecules had similar distributions. Small shifts were observed for some calculated descriptors when comparing correctly classified and misclassified compound distributions, but it was not possible to extract specific characteristics that determine errors in ML-QSPR models. We hypothesize that due to the multivariate nature of ML, especially deep learning models, it is unlikely that a small subset of features / characteristics explain model errors.

8. Regrettably, while the results are interesting, the quality and visual allure of the figures don't meet expectations. There are inconsistencies in font sizes and types, varying styles in the axes (e.g., ticks vs. non-ticks, unlabeled axes), and issues with readability (for instance, Figure 1 has minuscule fonts). Moreover, the panels in Figure 5 should be consolidated into a single figure block.

Response: We have regenerated the manuscript's figures to improve quality and consistency. We appreciate the reviewer's suggestions.

9. The UMAP figure is insufficiently explained and appears pixelated in the PDF version. Which descriptors or fingerprints were utilized?

Response: We have increased the quality of the UMAP figure and specified the molecular representation utilized (i.e. MACCS keys) in the main text.

10. In line 291, "Figure 3a" should be corrected to "Figure 4a."

Response: Figures have been reordered and should be correct now.

On the whole, this research conveys an important and positive message to the TPD community. However, not all modelers, particularly those in academia, will have the opportunity to work with such rich and extensive datasets. Therefore, as mentioned earlier, it would be good if the authors made an effort to share the computational resources presented here with the wider community in some form.

Response: We hope that the additions in terms of source data availability and surrogate data sets (mentioned above) facilitates advances in the ML-based QSPR models for TPD applications.

Reviewer #3 (Remarks to the Author):

Thank you for allowing me to review the manuscript by Peteani et al., "Are machine learning-based QSPR models applicable to targeted protein degraders?" (NCOMMS-23-44731).

Comments:

The authors constructed deep learning-based prediction models to predict the important ADME

properties focused on glues and heterobifunctional TPDs. ADME properties prediction for TPDs succeeded, and more effort was put into transfer learning. The authors also showed the difficulty of predicting bi-functionals. It would be informative and contribute to future PK studies to examine the various methods to predict ADME properties with large-scale data. Their objectives and results have been well demonstrated. Although it is recommended to add some additional analyses to prove the conclusion more clearly, I recommend publication after the minor revisions noted below.

Response: Thanks for the good feedback and time for reviewing our manuscript.

Recommendation:

Minor comment:

1. Regrettably, analyses using in-house company data are generally not fully disclosed in the data set. I would be glad if you could explain in as much detail as possible the distribution of the data set and their chemical space. Please visualize the difference in chemical space and molecular weight distribution between the training set and the test set in each compound group, such as all, bfx, and glues by UMAP. The degree of similarity between past and recent data is essential information that directly affects accuracy.

Response: We have now included more about the data distribution and its chemical space. In particular, the distributions of additional properties of the different chemical categories, including hydrogen bond acceptors (HBA), donors (HBD), topological polar surface area (TPSA), calculated LogP (cLogP), and number of rotational bonds are reported as Figure 2a. The differences in these properties between the training and test sets are shown in Figure S2.

2. The data extracted from the 25 assays differ in number for each parameter. How did you treat the data for which no numerical values were present in the dataset for these multi-tasking studies? Please add an explanation in the Materials and Methods section. Also, have you considered how the proportion of data for which no numerical values are present affects the results?

Response: We have included information about this in pg.30: *To enable MT model training with sparse labels, a masked loss function was utilized and missing values were not considered for backpropagation (ACS Omega 2019, 4, 2, 4367–4375). Previous investigations indicated that especially for sparse experimental data, MT learning can provide an advantage compared to single-task models (ACS Omega 2018 3 (9), 12033-12040).*

3. Please reconsider Figure 2; the difference between Strategy 1 and 2 cannot be taken from Figure 2.

Response: We have modified the coloring of Figure 5 (previously, Figure 2). The grey color which was identifying the TPD compounds) was removed to emphasize the difference between Strategy 1 and 2.

4. p18, section 3.2, Please describe in more detail how you constructed the baseline predictor.

Response: To further describe the baseline predictor, the following sentence has been included (pg.14): *For all test compounds, the baseline model gave a constant prediction value which corresponded to the mean property value in the training set.*

REVIEWERS' COMMENTS

Reviewer #1 (Remarks to the Author):

This report is the first comprehensive evaluation of ML for the prediction of ADME and physiochemical properties of TPD molecules, including heterobifunctional and molecular glue submodalities. Several areas for improvement of the manuscript were noted by reviewers in the areas of broad overstatements, typos, lack of experimental details, insufficient information on the chemical data sets, figure/graph quality, and lack of code/data availability. The authors did an excellent job addressing the reviewers comments. In particular, the addition of the new Figure 2 showing the distribution of properties for the 3 groups of molecules and the addition of the analysis of public surrogate data and ML software will provide important reference data and tools for the pharmaceutical industry to help advance the field. I support the acceptance and publication of the manuscript in its current state without further revision.

Reviewer #1 (Remarks on code availability):

I do not have the expertise to review the code.

Reviewer #2 (Remarks to the Author):

Authors have addressed my comments satisfactorily. While they were unable to share data or models, they provide "surrogate datasets" of significant scale. I think this is a good solution.

Having said that, it would be great to have models deployed as an online service maintained by Novartis (I understand that models cannot be shared due to risk of reverse engineering). This is not a must, in my opinion, and I defer to the editors and other reviewers to reach a final decision.